# IT'S ABOUT TIME: TEMPORAL REFERENCES IN EMERGENT COMMUNICATION

## ABSTRACT

As humans, we use linguistic elements referencing time, such as "before" or "tomorrow", to easily share past experiences and future predictions. While temporal aspects of the language have been considered in computational linguistics, no such exploration has been done within the field of emergent communication. We research this gap, providing the first reported temporal vocabulary within emergent communication literature. Our experimental analysis shows that the ability to process temporal relationships is sufficient for the natural emergence of temporal references, and that no additional losses are necessary. Our readily transferable architectural insights provide the basis for the incorporation of temporal referencing into other emergent communication environments.

## 1 INTRODUCTION

How will autonomous agents communicate? What will the resulting language look like if it's left to them to design their own complex languages? These questions have been examined through the lens of Emergent Communication (EC) (Lazaridou and Baroni, 2020), where agents develop their language from scratch. The resulting language is usually tailored to the specific environment in which they have been trained, with the language reflecting the tasks the agents perform, the actions available to them and the other agents they interact with. These properties make the emergent language memory and bandwidth efficient, as the agents can optimise their vocabulary size and word length to their specific task, providing an advantage over a general, hand-crafted communication protocol.

Another aspect of the communication protocol which may benefit the agents is the ability to reference previous observations. For example, agents deployed in autonomous vehicles can share information about previously encountered obstacles or past traffic conditions. Agents working in finance can share their experiences of past trading operations, trading performance and past financial data. Agents tasked with monitoring cybersecurity could more easily share information about past incidents and attack patterns to prevent future threats. As environmental complexity is being scaled in emergent communication research (Chaabouni et al., 2022), temporal references will also benefit agents in settings where temporal relationships are embedded. One example is social deduction games (Brandizzi et al., 2021; Lipinski et al., 2022; Kopparapu et al., 2022), where referencing past events are expected to be key to winning strategies. Temporal references will also allow agents to develop more efficient methods of communication by assigning shorter messages to events which happen more often, similarly to Zipf's Law in human languages Zipf (1949). The temporal references, in conjunction with the general characteristics of emergent languages, will enhance the agent's bandwidth efficiency and task performance in a variety of situations.

While temporal aspects of the language have been considered in linguistics (Spronck and Casartelli, 2021), there is no recorded natural emergence of temporal references, nor any exploration of how they may emerge in communication among agents. Prior research in EC has investigated the influence of time as a pressure for protocol emergence (Kalinowska et al., 2022a;b), temporally separated tasks (Ossenkopf et al., 2019), or the effect of the amount of communication time on the emergent protocol (Lipinski et al., 2022). Most closely related is the work of Kang et al. (2020), where temporal relationships between episodes have been employed to optimise communication. The authors exploit similarity between time steps, where subsequent time steps do not differ significantly from the ones immediately preceding. This allows the messages between agents to be more succinct, by reducing the amount of redundant information transferred. Kang et al. (2020) look into the pragmatics of

the language, noting that through utilisation of temporal relationships, together with optimisation of reconstruction of speaker's state, the agents' performance improves. However, the temporal aspects of the language itself have not been explored.

We investigate the factors of temporal reference emergence, thus closing this gap. The main contribution of this work is the first reported temporal vocabulary developed by agents in a referential game. We analyse the incentives required for the development of temporal references in language through an environment we call the Temporal Referential Game (TRG) (Section 2.3). We show that, surprisingly, only the ability to process temporal relationships is needed for the agents to be able to understand and utilise temporal references (Section 2.4).

## 2 TEMPORAL REFERENTIAL GAMES

Our experimental setup is based on classic referential games (Lewis, 1969; Lazaridou et al., 2018). The commonly used agent architecture, as implemented in Kharitonov et al. (2019) and similarly in other works (Chaabouni et al., 2020; Taillandier et al., 2023; Bosc, 2022; Ueda and Washio, 2021), has two agents: a sender and a receiver. The sender begins the game by observing a target object, which could be represented by an image or a vector, and then generates a message. This message is passed to the receiver, along with the target object and a number of distractor objects. The receiver's task is to discern the target object from among the objects it observes, using the information contained in the message it receives. This exchange is repeated every episode.

We use referential games with attribute-value vectors to isolate and limit the external factors that could impact the performance of the agents. We do not use image-based representations of the objects to separate the performance of the agent from the training and performance of a vision network, and to reduce the computational requirements of our experiments. Additionally, the output of a vision layer can be considered as a representation of the object attributes, which could be approximated by the vectors used in our setup instead.

This common approach from EC (Kharitonov et al., 2019; Chaabouni et al., 2020; Ueda et al., 2022), allows us to use a well-known test bed to probe the more complex temporal properties of the emergent language. By using a simple referential game, and removing extraneous modules, our findings are more generalisable and transferable to other settings.

### 2.1 DEFINITIONS

In referential games, agents need to identify objects from an *object space* $V$, which appear to them as attribute-value vectors $\boldsymbol{x} \in V$. To define the *object space* $V$, we first define the *value space* of all possible attribute values as $S_{val} = \{0, 1, 2 \ldots N_{val}\}$ where $N_{val}$ is the *number of values*. The value space represents the variations each object *attribute* can have. The *object space* is defined as $V = V_1 \times \cdots \times V_N = \{(a_1, \ldots, a_{N_{att}}) \mid a_i \in V_i \text{ for every } i \in \{1, \ldots, N_{att}\}\}$, where $N_{att}$ is the *number of attributes* of an object.

To give intuition to the notion of attributes and values, consider that the object shown to the sender is an abstraction of an image of a circle. The attributes of the circle could include whether the line is dashed, the colour of the line, or the colour of the background. The values are the variations of these attributes. In our example, a value of the background colour could be black, blue, or red. For example, to represent a blue, solid line circle on a red background a vector such as [blue, solid, red] could be used, which could also be represented as an integer vector, for example $[2, 1, 3]$.

The characters available to the agents (*i.e.*, *the symbol space*) is $\omega = \{0, 1, 2 \ldots N_{vocab} - 1\}$ where $N_{vocab}$ is the *vocabulary size*. The *message space*, or the space that all messages must belong to, is defined as $\xi = \omega_1 \times \cdots \times \omega_L = \{(c_1, \ldots, c_L) \mid c_i \in \omega_i \text{ for every } i \in \{1, \ldots, L\}\}$, where $L$ is the maximum message length.

Combining the message and object space, the agents' language is defined as a mapping from the objects in $V$ to messages in $\xi$. Finally, the exchange history, representing all messages and objects that the agents have sent/seen so far, is defined as a sequence $\tau = \{(\boldsymbol{m}_n, \boldsymbol{x}_n)\}_{n \in \{1, \ldots, t\}}$ such that $\forall n, \boldsymbol{m}_n \in \xi \wedge \boldsymbol{x}_n \in V$, with $t$ signifying the episode of the last exchange. Agent communication is defined as agents using this language to convey information about the observed object.

## 2.2 TEMPORAL LOGIC

We use temporal logic to formally define the behaviour of our environment, as well as an analogue for how our agents communicate. To achieve this, we employ a form of Linear Temporal Logic (LTL) (Pnueli, 1977) called Past LTL (PLTL) (Lichtenstein et al., 1985), bringing common terminology from the logic domains into the field of emergent communication.

LTL focuses on the connection between future and present propositions, defining operators such as "next" $\bigcirc$, indicating that a given predicate or event will be true in the next step. The LTL operators can then be extended to include the temporal relationship with propositions in the past, creating PLTL. PLTL defines the operator "previously" $\ominus$, corresponding to the LTL operator of "next" $\bigcirc$.

The "previously" PLTL operator must satisfy Equation (1), using the definitions from Maler et al. (2008), where $\sigma$ refers to a behaviour of a system (the message sent by an agent), at the time $t$ to the time that event has occurred (when the message was sent), and $\phi$ signifies a property (the object seen by the agent).

$$(\sigma, t) \models \ominus\phi \leftrightarrow (\sigma, t-1) \models \phi \tag{1}$$

Additionally, the shorthand notation of $\ominus^n$ is used, signifying that the $\ominus$ operator is applied $n$ times, where $n$ refers to the number of episodes back. For instance, $\ominus^4\phi \leftrightarrow \ominus\ominus\ominus\ominus \phi$.

## 2.3 TEMPORAL REFERENTIAL GAMES

**Our temporal version** of the referential games (Lewis, 1969; Lazaridou et al., 2017) is based on the "previously" ($\ominus$) PLTL operator. [1] At every game step $s_t$, the sender agent is presented with an input object vector $\boldsymbol{x}$ generated by the function $X(t, c, h_v)$, with a random *chance* parameter $c$, the *previous horizon* value $h_v$, and the current episode $t$. [2]

$$X(t, c, h_v) = \begin{cases} \boldsymbol{x} & c = 0 \\ \ominus^{h_v}\boldsymbol{x} = \tau_{t-h_v} & c = 1 \end{cases} \tag{2}$$

The *previous horizon* value is uniformly sampled, taking the value of any integer in the range $[1, h]$, where $h$ is the *previous horizon* hyperparameter. We sample the *previous horizon* value to allow agents to develop temporal references of varying temporal horizons, instead of fixing the parameter each run. The function $X(t, c, h_v)$ selects a target object to be presented to the sender using Equation (2), either generating a new random target object or using the old target object. This choice is facilitated using the *chance* parameter $c$, which is sampled from a Bernoulli distribution, with $p = 0.5$. If $c = 1$ a previous target object is used, and if $c = 0$ a new target object is generated. Both $c$ and $h_v$ are sampled every time a target object is generated.

For example, consider episode $t = 4$, and the sampled parameters are $c = 1$ and $h_v = 2$. Suppose the agent has observed the following targets: $[\boldsymbol{a}, \boldsymbol{b}, \boldsymbol{c}]$. Given that $c = 1$, further to Equation (2), the $\ominus^2$ ($\ominus^{h_v}$) target is chosen. The target sequence becomes $[\boldsymbol{a}, \boldsymbol{b}, \boldsymbol{c}, \boldsymbol{b}]$, with the target $\boldsymbol{b}$ being repeated, as it was the second to last target. Now suppose that $c$ was sampled to be $c = 0$ instead. Further to Equation (2), a random target $\boldsymbol{x}$ is generated, from $\boldsymbol{x} \in V$. The target sequence becomes $[\boldsymbol{a}, \boldsymbol{b}, \boldsymbol{c}, \boldsymbol{x}]$.

This behaviour describes the environment "TRG Previous", which represents the base variant of temporal referential games, where targets are randomly generated with a 50% chance of repetition. The "TRG Hard" variant is also used, which is a temporal referential game with the same 50% chance of a repetition, but where targets only differ in a single attribute when compared to the distractors. "TRG Hard" tests whether temporal referencing improves performance in environments where highly similar target repetitions are common. [3]

The agents are also trained and evaluated in the "RG Classic" environment, which represents the classic referential game (Lewis, 1969; Lazaridou et al., 2017), where targets are randomly generated,

---

[1]Code is available on Anonymous GitHub

[2]Additional details are available in the Appendix Cl.

[3]Measurements of amounts of repetitions in each environment are provided in Appendix B.

and "RG Hard", which is our more difficult version of the referential games, where the target and distractors only differ in a single attribute. The "RG Classic" environment establishes a reference performance for the agents, while "RG Hard" determines whether temporal references enhance performance in an environment where targets are harder to differentiate.

Additionally, two more environments are used — "Always Same" and "Never Same". Their purpose is to verify whether the messages that would be identified as temporal references are correctly labelled. The "Always Same" environment sequentially repeats each target from a uniformly sampled subset of all possible targets ten times[4]. With each target repeating ten times, we verify that the messages used are consistently; *i.e.*, if the agents use temporal messaging. The "Never Same" never repeats a target and goes through a subset of all possible targets in order. The "Never Same" environment is used to verify if the same messages are used for other purposes than to purely indicate that the targets are the same. In both environments, the dataset only repeats the target object, while the distractor objects are randomly generated for each object set. Sample inputs and expected outputs for these environments are provided in Appendix B.1.

## 2.4 AGENT ARCHITECTURE

Both the sender and the receiver agents are usually built around a single recurrent neural network Kharitonov et al. (2019), such as an LSTM (Hochreiter and Schmidhuber, 1997) or a GRU (Cho et al., 2014). These networks are supplied with a representation of the objects and, in the case of the receiver, a message. These representations are obtained using fully connected layers, with the message being either discrete or a distribution of character probabilities in the case of Gumbel-Softmax.

We introduce **a second LSTM** module in both the sender and receiver networks, *cf.*, Figure 1. In other approaches (Kharitonov et al., 2019; Chaabouni et al., 2019; Auersperger and Pecina, 2022), the sender's LSTM receives each target and distractor set individually and computes the message, processing each object separately. Instead, our additional LSTM is batched with a sequence over the whole training input, similar to the sequential learning of language in humans (Christiansen and Kirby, 2003). By including this sequential LSTM, the sender and the receiver are able to develop a more temporally focused understanding. We conjecture that this ability to process temporal relationships allows them to represent the whole object sequence within the LSTM hidden state. Since it does not require reward shaping approaches or architectures specifically designed for referential games, this addition is also a scalable and general approach to allowing temporal references to develop.

To give intuition to the sequential LSTM, assume the sender LSTM expects an input of the form $[batch\_size, seq\_len, object\_attributes]$. Let $batch\_size$ take the common value of 128, and let the $object\_attributes$, or $N_{att}$, be equal to 6. We can then create a batch of shape $[128, 1, 6]$, obtaining 128 objects of size 6, with sequence length one (Kharitonov et al., 2019). The sequential LSTM instead receives a batch of shape $[1, 128, 6]$, or a sequence of 128 objects of size 6. This allows the sequential LSTM to process all objects one after another to create temporal understanding.

The hidden states gathered from both sender LSTMs are combined using an element-wise multiplication, which returns the combined state. The result is the initial hidden state for the message generation LSTM. For message generation, the same method is followed as used in previous work (Kharitonov et al., 2019), and messages are generated character by character, using the Gumbel-Softmax trick (Jang et al., 2017).

These messages are then passed to the receiver, an overview of which is shown in Figure 1b. The receiver's architecture contains an object embedding linear layer and a message processing LSTM, similar to the most commonly used architectures (Kharitonov et al., 2019). We additionally employ the temporal prediction layer and the sequential LSTM. First, a hidden state is computed for each message by the regularly batched LSTM. Then, the sequential LSTM processes each of the regularly batched LSTM's hidden states to build a temporal understanding of the sender's messages. This output is combined with the output of the object embedding linear layer to create the referential game object prediction. The combined information from both LSTMs and the object is also used in the temporal prediction layer, which allows the agent to signify whether an object is the same as a previously seen object, up to the previous horizon $h$. This is implemented as a single linear layer, which outputs the temporal label prediction. The temporal label used in this loss function only

---

[4]A subset is used as the target space grows exponentially with the number of attributes and values.

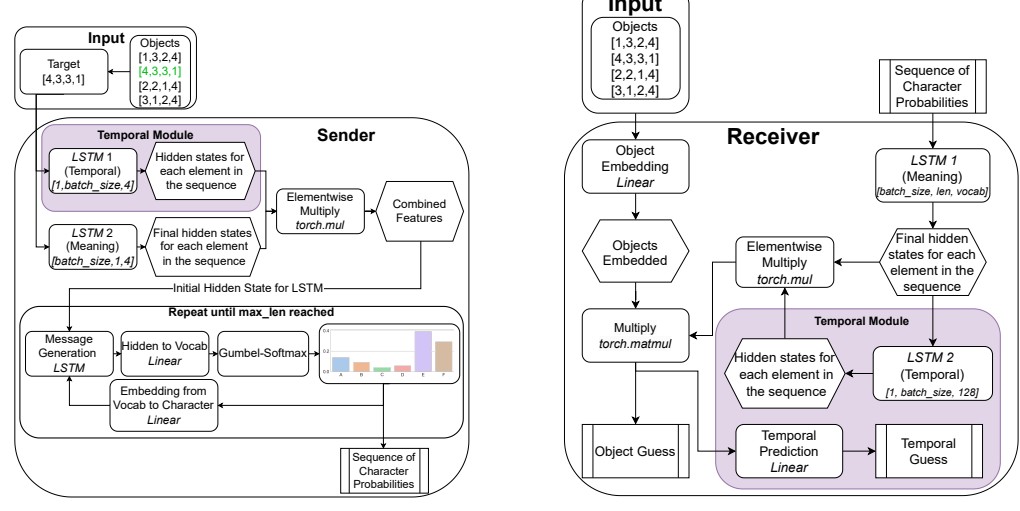

(a) The sender architecture.        (b) The receiver architecture.

Figure 1: The sender and receiver architectures, with the temporal modules highlighted in purple.

considers the previous horizon $h$; otherwise, it defaults to $0$. For example, assume an object has been repeated in the current episode and has last appeared $5$ episodes ago. If the previous horizon $h$ is $8$, the label assigned to this object would be $5$, as $5$ past episodes are still within the horizon, *i.e.*, $5 \leq h$. However, if $h$ is $4$, the label would be $0$, as the episode lies outside the previous horizon, *i.e.*, $4 \geq h$.

This predictive ability is combined with an additional term in the loss function, which together form a **temporal prediction loss**. The agents' loss function can be formulated as $L_t = L_{rg} + L_{tp}$. The $L_{rg}$ component is the referential game loss between the receiver guess and the sender target label, using cross entropy. $L_{tp}$ is the temporal prediction loss, which is implemented using cross entropy between the labels of when an object has last appeared, and the receiver's prediction of that label. Agents that include this loss perform an additional task, which corresponds to correctly identifying which two outputs are the same. The goal of this loss is to improve the likelihood of an agent developing temporal references by increasing the focus on these relationships. Analysis of how the presence of this explicit loss impacts the development of temporal references is provided in Section 3.

## 3 TEMPORALITY EXPERIMENTS

### 3.1 TEMPORALITY METRIC

We propose a metric, denoted as $M_{\ominus^n}$, which measures how often a given message has been used as the "previous" operator in prior communication. Given a sequence of objects shown to the sender and messages sent to the receiver $\tau$, it checks when an object has been repeated within a given horizon $h_v$, and records the corresponding message sent to describe that object.

Let $C_{\boldsymbol{m}\ominus^n}$ count the times the message $\boldsymbol{m}$ has been sent together with a repeated object for $h_v = n$:

$$C_{\boldsymbol{m}\ominus^n} = \sum_{j=1}^{t} \mathbb{I}(\boldsymbol{m}_j = \boldsymbol{m} \wedge \text{objectSame}(\boldsymbol{x}_j, n)) \tag{3}$$

where $\mathbb{I}(\cdot)$ is the indicator function that returns $1$ if the condition is true and $0$ otherwise, and $\text{objectSame}(\boldsymbol{x}_j, n)$ is a function that evaluates to true if the object $\boldsymbol{x}_j$ is the same as the object $n$ episodes ago.

Let $C_{\boldsymbol{m}}\text{total}$ denote the total count of times the message $\boldsymbol{m}$ has been used:

$$C_{\boldsymbol{m}}\text{total} = \sum_{j=1}^{t} \mathbb{I}(\boldsymbol{m}_j = \boldsymbol{m}) \tag{4}$$

where $\mathbb{I}(\cdot)$ is an indicator function selecting the message $\boldsymbol{m}$ in the exchange history $\tau$.

The percentage of previous messages that are the same as $\boldsymbol{m}$ can then be calculated using $M_{\ominus^n}(\boldsymbol{m})$:

$$M_{\ominus^n}(\boldsymbol{m}) = \frac{C_{\boldsymbol{m}\ominus^n}}{C_{\boldsymbol{m}}\text{total}} \times 100 \tag{5}$$

To give intuition to this metric, its objective is to measure if a message is used similarly to the sentence "The car I can see is the same colour as the one mentioned two sentences ago", *i.e.*, if the message can give reference to a previous episode. More formally, assume a target object sequence of $[\boldsymbol{x}, \boldsymbol{y}, \boldsymbol{z}, \boldsymbol{y}, \boldsymbol{y}, \boldsymbol{y}, \boldsymbol{x}]$. Each vector — $\boldsymbol{x}$,$\boldsymbol{y}$,$\boldsymbol{z}$ — represents an object belonging to the same arbitrary $V$. In this example, there is only one object repeating: $\boldsymbol{y}$. We can then consider three message sequences: $[\boldsymbol{m}_1, \boldsymbol{m}_2, \boldsymbol{m}_3, \boldsymbol{m}_2, \boldsymbol{m}_4, \boldsymbol{m}_4, \boldsymbol{m}_1]$, $[\boldsymbol{m}_1, \boldsymbol{m}_2, \boldsymbol{m}_3, \boldsymbol{m}_4, \boldsymbol{m}_4, \boldsymbol{m}_4, \boldsymbol{m}_1]$ and $[\boldsymbol{m}_1, \boldsymbol{m}_2, \boldsymbol{m}_3, \boldsymbol{m}_2, \boldsymbol{m}_2, \boldsymbol{m}_2, \boldsymbol{m}_1]$, with each $\boldsymbol{m}_n$ belonging to the same arbitrary $\xi$. Given these sequences, we can calculate our metric for $\ominus^1$.

There are two repetitions in the sequence of objects: the second and third $\boldsymbol{x}$ following the sequence of $[\boldsymbol{x}, \boldsymbol{y}, \boldsymbol{z}, \boldsymbol{y}]$. In the first example message sequence, for both of the repetitions, the message $\boldsymbol{m}_4$ has been sent and so $C_{\boldsymbol{m}_4}\ominus^1 = 2$. The total use of $\boldsymbol{m}_4$ is $C_{\boldsymbol{m}_4}\text{total} = 2$. Calculating the metric $M_{\ominus^1}(\boldsymbol{m}_4) = 2/2 \times 100 = 100\%$ gives 100% for the use of $\boldsymbol{m}_4$ as a $\ominus^1$ operator. The result of 100% indicates that this message is used exclusively as a $\ominus^1$ operator.

In the second message sequence, $\boldsymbol{m}_4$ has also been used for the initial observation of the object. This means that $C_{\boldsymbol{m}_4}\text{total} = 3$, while $C_{\boldsymbol{m}_4}\ominus^1 = 2$. We can calculate $M_{\ominus^1}(\boldsymbol{m}_4) = 2/3 \times 100 = 66\%$, which shows the message being used as $\ominus^1$ 66% of the time.

Lastly, the simplest case of a message describing an object exactly. Following the previous examples, $C_{\boldsymbol{m}_2}\text{total} = 4$, with $C_{\boldsymbol{m}_2}\ominus^1 = 2$. This message would then be classed as 50% $\ominus^1$ usage, $M_{\ominus^1}(\boldsymbol{m}_2) = 2/4 \times 100 = 50\%$. A non-100% result indicates that the message is not used exclusively as a $\ominus^1$ operator.

## 3.2 AGENT TRAINING

The following architectures are evaluated:

***Non-Temporal-NL*** (*NL* meaning *No-Loss*) Same as regular emergent communication agents, which is used as a baseline for comparison;
***Non-Temporal*** Same as regular emergent communication agents, **but** with temporal prediction loss;
***Temporal-NL*** Includes the sequential LSTM, but **not** the temporal prediction loss; and
***Temporal*** Includes both the sequential LSTM **and** the temporal prediction loss.

The agents that include the temporal prediction loss (*i.e.*, *Temporal* and *Non-Temporal*) have an explicit reward to develop temporal understanding. There is no additional pressure to develop temporal references for agents that do not include the temporal prediction loss (*i.e.*, *Temporal-NL* and *Non-Temporal-NL*), except for the possibility of increased performance on the referential task.

We hypothesise that *Non-Temporal*, *Temporal-NL* and *Temporal* agents will develop temporal references, with the *Temporal* agents more likely to do so, given their incentive is higher. We define the development of temporal references as the appearance of messages which reach 100% on our $M_{\ominus^n}$ metric. We use the cut-off of 100% to ensure we only report messages used consistently as temporal references, ensuring that any positive results are not a result of chance repetitions. This means that we should see our $M_{\ominus^n}$ metric reach 100% for all these agents, but not for the *Non-Temporal-NL* agents.

All agent types were trained for the same number of epochs and on the same environments during each run. Evaluation of the agents is performed after the training has finished. Each agent pair is assessed in the six different environments: "Always Same", "Never Same", "RG Classic", "RG Hard", "TRG Previous" and "TRG Hard". The target objects are uniformly sampled from the object space $V$ in all environments.

Each possible configuration was run ten times, with randomised seeds between runs for both the agents and the datasets. Each agent pair was then evaluated in six different environments. Appendix A provides further details.

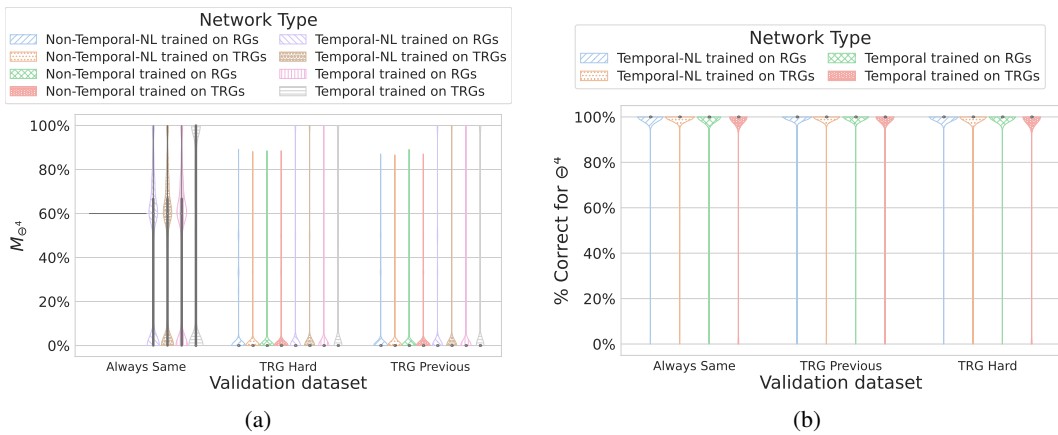

Figure 2: The $M_{\ominus^4}$ metric (a) and correctness of messages (b) used as the $\ominus^4$ operator.

## 3.3 TEMPORALITY ANALYSIS

Figure 2a illustrates the $M_{\ominus^4}$ metric values (referring to an observation four messages in the past) of all agent types over the evaluation environments (*cf.*, Sections 3.1 and 3.2), where $M_{\ominus^4} \geq 0\%$ [5]. "TRGs" refers to "TRG Previous", and "RGs" refers to "RG Classic". Figure 2a indicates that the temporally focused processing of the input data makes the agents predetermined to develop temporal references. Only the networks that have the sequential LSTM, *i.e.*, *Temporal* and *Temporal-NL*, are capable of producing temporal references. Conversely, temporal references emerge in both *Temporal* and *Temporal-NL* networks, regardless of the training dataset. This shows that even in a regular environment, without additional pressures, temporal references are advantageous. No messages in the *Non-Temporal* or *Non-Temporal-NL* architectures are used $100\%$ of the time for $\ominus^4$, irrespective of the dataset they have been trained on. This demonstrates that the temporal prediction loss is not enough, and that a sequential LSTM module is the key factor to the emergence of temporal references.

Figure 2b shows that messages that are used for $\ominus^4$ have a high chance of being correct, with most averaging above $90\%$ **correctness**. **Correctness** refers to whether the receiver agent correctly guessed the target object after receiving the message. As expected, no *Non-Temporal* networks appear in Figure 2b because they learn no temporally specialised messages, and so no messages are used as $\ominus^4$.

Analysing the development of temporal references, we observe the emergence of messages being used by the agents to describe the previous $h_v = 4$ episodes. As an example of such behaviour, in one of the runs where the agents were trained in the *Temporal* configuration, the message $[25, 6, 9, 3, 2]$ was consistently used as a $\ominus^1$ operator. When the agents were evaluated in the "Always Same" environment, they used this message only when the target objects were repeating, while also being used exclusively for twelve distinct objects. For a total of 10 repetitions of each object, this message was utilised nine times, indicating that the only time a different message was sent was when the object appeared for the first time. For example, when the object $[4, 2, 3, 6, 5, 8, 8, 4]$ appeared for the first time, a message $[25, 6, 17, 9, 9]$ was sent, and subsequently the temporal message was used. This shows that temporal messages aid generalisation. A message that has been developed in a different training environment, in this case "TRG Previous", can be subsequently used during evaluation, even if the targets are not shared between the two environments.

The distribution of all messages as compared to their $M_{\ominus^4}$ value is shown in Figure 3a. Most messages are used only in the context of the current observations, with both *Temporal* and *Temporal-NL* networks using a more specialised subset of messages to refer to the temporal relationships. Only *Temporal* and *Temporal-NL* variants develop messages that reach $100\%$ on the $M_{\ominus^4}$ metric. The distribution also suggests that these messages could be a more efficient way of describing objects, as the number of temporal messages is relatively small. Since only a small number of messages are needed for the temporal references, they could be used more frequently. This message specialisation,

---

[5]We choose the value of 4 arbitrarily, to lie in the middle of our explored range of $h$. We provide more detailed analysis from $h_v = 1$ to $h_v = 8$ in Appendix E.

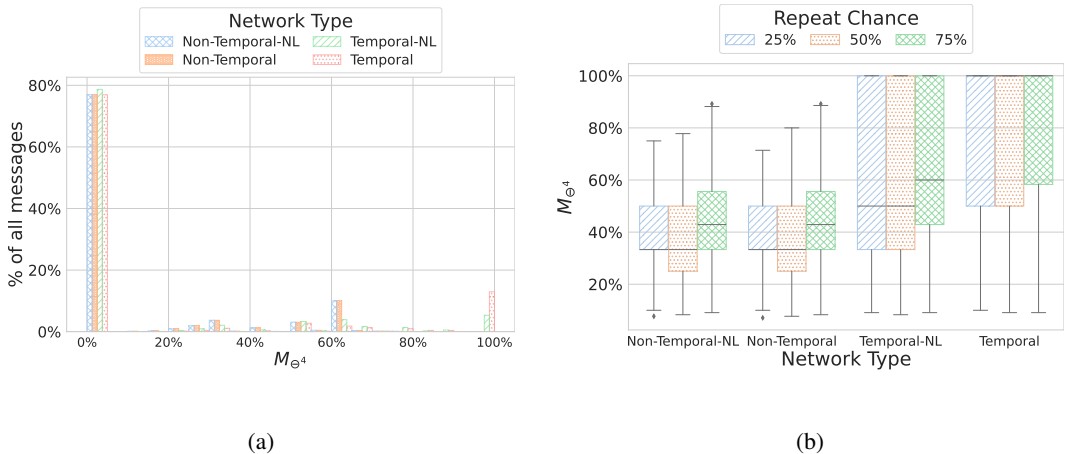

(a)                                                                         (b)

Figure 3: Usage of messages compared to their $M_{\ominus^4}$ value (a), and of the $M_{\ominus^4}$ value when varying the network type and the chance of repetition $p$ (b).

Table 1: Percentage of networks that develop temporal messages.

| Network Type | Loss Type | Percentage |
|---|---|---|
| Non-Temporal | Non-Temporal | $0\%$ |
| Non-Temporal | Temporal | $0\%$ |
| Temporal | Non-Temporal | $98.66\%$ |
| Temporal | Temporal | $97\%$ |

combined with a linguistic parsimony pressure (Rita et al., 2020), could lead to a more efficient way of describing an object, as sending the object properties requires more bandwidth than sending only the time step the object last appeared.

The percentage of networks that develop temporal messaging is shown in Table 1. The percentages shown are absolute values, calculated by taking the total number of runs and checking whether at least one message has reached $M_{\ominus^n} = 100\%$ for each run. That number of runs is divided by the total number of runs of the corresponding configuration to arrive at the quantities in Table 1.

In Table 1, both *Temporal* and *Temporal-NL* network variants reach over $95\%$ of runs that have converged to a strategy which uses at least one message as the $\ominus^n$ operator. In contrast, the *Non-Temporal* and *Non-Temporal-NL* networks never achieve such a distinction. Additionally, some runs have not converged to a temporal strategy in the case of both *Temporal* network variants. However, these experiments account for only $3\%$ of the total number of runs. These results further show that the ability to build a temporal understanding of the input data is the deciding factor in the emergence of temporal references. Only networks that include the sequential LSTM converge to strategies that include such references.

To thoroughly investigate this result, we analyse the impact of the network size on the development of temporal references. We evaluate agents with just the temporal module, removing the Meaning LSTM 2 for the sender and the Meaning LSTM 1 for the receiver, matching the number of parameters as observed in the base agent (Section 2.4). The agents with just the temporal module still develop temporal references, but perform worse on the referential task, achieving lower accuracy. Therefore, they are omitted from the comparisons.

## 4 DISCUSSION

When and how can temporal references emerge? We posit that our addition of the sequential LSTM to the agents is key to allowing them to develop the ability to communicate about time. The fundamental factor in the emergence of temporal references is whether the agents can look into the past, which the

sequential LSTM allows them to do. Our results support this, showing the inclusion of this module is *sufficient* for agents to form temporal references.

In Figure 3b, we verify that by increasing the number of repetitions in a dataset, the use of temporal messages increases. As we increase the repetition chance, the percentage of messages that are used for $\ominus^n$ increases for all agent variants. On average, *Non-Temporal* and *Non-Temporal-NL* networks demonstrate the same chance of using a message for $\ominus^n$ as the dataset repetition chance. This means that while the percentage increases, it is only due to the increase in the repetition chance. If a dataset contains 75% repetitions, on average, each message will be used as an accidental $\ominus^n$ 75% of the time. For example, if the language does not have temporal references and uses a given message to describe an object, this message will be repeated every time this object appears. This means that for every repetition, the message could be seen as a message indicating a *previous* episode, whereas in reality, it is just a description of the object. In contrast to *Non-Temporal* and *Non-Temporal-NL* networks, for *Temporal* and *Temporal-NL* networks, the average percentage does reach 100%. This means that messages the agents designate for $\ominus^n$ are used more often than the repetition chance.

Our results also indicate that the only pressures required for temporal messages to emerge are *implicit*, and that no explicit pressures are required. We show that the incentives are already present in datasets that are *not* altered to increase the number of repetitions occurring. Temporal references therefore emerge naturally, as long as the agents are able to build a temporal understanding of the data, such as with the sequential LSTM used in our work. This ease of transfer of our insights allows temporal references to emerge in any emergent communication settings. This could allow for greater bandwidth efficiency by allowing agents to use shorter messages for events that happen often, especially when combined with other linguistic parsimony approaches (Rita et al., 2020; Chaabouni et al., 2019).

The emergence of temporal references only through architectural changes could also point towards additional insights in terms of modelling human language evolution using EC (Galke et al., 2022). Our sequential LSTM approach to the emergence of temporal references could be viewed as analogous to sequential learning in natural language (Christiansen and Kirby, 2003), as we learn to encode and represent elements in temporal sequences.

## 5    LIMITATIONS

It's possible for the agent to send a unique message that describes the time of the object's appearance, rather than sending a message it has used before. Our metric would then incorrectly identify the training run as having no temporal references, given that all messages would be unique. This would require the agents to develop a very large vocabulary, creating a unique message for every object repetition. We do not, however, observe this happening in our training runs, as our agents' vocabulary contains at most 4k messages over the whole training run. This is significantly smaller than the number of repeated objects, which in the case of the 50% repetition dataset would be 10k. Using the pigeonhole principle, we can conclude that they do not create a unique message for each repetition. We consider that this limitation is related to the issues with most compositionality metrics in EC. While most compositionality metrics measure trivial compositionality (Chaabouni et al., 2020; Ueda et al., 2022; Perkins, 2021), our metric would be akin to measuring trivial temporality.

## 6    CONCLUSION

Discussing past observations is vital to communication, saving bandwidth by avoiding repeating information and allowing for easier experience sharing. We investigate the emergence of temporal references, addressing the fundamental questions of *when* and *how* they can develop. We present an environment to probe how agents might create such references. By testing environmental pressures, employing multiple network architectures, and incorporating a temporal referencing reward, we analyse the mechanisms underlying the formation of temporal references.

We perform a comparison of a conventional agent architecture with an architecture featuring the ability to understand temporal relationships in the data. We show that this change is sufficient for temporal references to emerge, finding the additional explicit incentive of a temporal prediction loss to be unnecessary. The ability to process observations temporally, combined with implicit pressures from the environment, allows temporal references to emerge naturally.

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

# A  TRAINING DETAILS

Our agents were trained using PyTorch Lightning (Falcon and The PyTorch Lightning Team, 2019) using the Adam optimizer (Kingma and Ba, 2015), with experiment tracking done via Weights & Biases (Biewald, 2020). We provide our grid search parameters per network and per training environment in Table 2. We ran a manual grid search over these parameters for each network and training dataset combination, where the networks were *Non-Temporal*, *Non-Temporal-NL*, *Temporal*, and *Temporal-NL*, and the training datasets were Classic Referential Games or Temporal Referential Games. Each trained network was then evaluated on the six available environments: Always Same, Never Same, Classic Referential Games, Temporal Referential Games, Hard Classic Referential Games, and Hard Temporal Referential Games. Running the grid search for one iteration, with the value of repetition chance fixed, took approximately 28 hours, using the compute resources in Table 3.

Table 2: Grid Search Parameters

| Parameter | Value |
|---|---|
| Epochs | [600] |
| Optimizer | Adam |
| Learning Rate $\alpha$ | 0.001 |
| Number of Objects in Dataset | [20 000] |
| Number of Distractors | [10] |
| Number of Attributes $N_{att}$ | [8] |
| Number of Values $N_{val}$ | [8] |
| Length Penalty | [0] |
| Maximum Message Length $L$ | [5] |
| Vocabulary Size $N_{vocab}$ | [26] |
| Repetition Chance ($p$) | [0.25, 0.5, 0.75] |
| Previous Horizon $h$ | [8] |
| Sender Embedding Size | [128] |
| Sender Meaning LSTM Hidden Size | [128] |
| Sender Temporal LSTM Hidden Size | [128] |
| Sender Message LSTM Hidden Size | [128] |
| Receiver LSTM+Linear Hidden Size | [128] |
| Gumbel-Softmax Temperature | [1.0] |

Table 3: Compute Resources

| Resource | Quantity |
|---|---|
| CPU Cores (Intel(R) Xeon(R) Silver 4216 × 2) | 20 |
| GPUs (NVIDIA Quadro RTX8000) | 1 |
| Wall Time | 28hrs |

# B  DATASETS DETAILS

In Figure 4, we analyse our datasets, using the parameters as specified in Appendix A, for the number of repetitions that occur. When the temporal dataset repetition chance is set to 50%, the datasets, predictably, oscillate around 50% of repeating targets. Generating the targets randomly yields a miniscule fraction of repetitions of less than 1%, as we can see in Figure 4, for the Classic and Hard referential games.

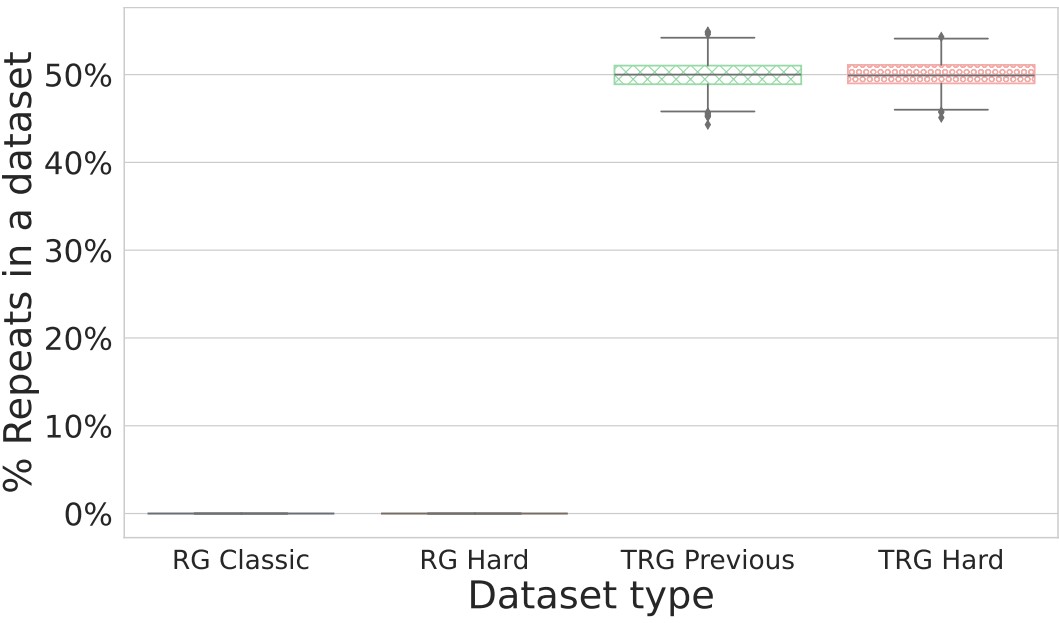

Figure 4: Number of target repetitions per dataset. Regular referential games datasets very rarely encounter target repetitions. This data is an average over 1000 seeds per environment.

## B.1  TEST ENVIRONMENTS

Both "Always Same" and "Never Same" environments act as sanity checks for our results.

We provide example inputs and outputs for both environments in Table 4 and Table 5. We use single-attribute objects and messages for clarity.

For the "Always Same" environment, in the case of the agent using temporal references, we may also see other messages instead of the message 4, as we have observed that there are more than one message used as previously. We always expect to see at least 90% of usage as previously for this environment. However, for agents that learn temporal referencing strategies, we would expect the usage to reach 100%.

For the "Never Same" environment, we expect to see no temporal references being identified. Any identification of temporal references in the Never Same environment would indicate an issue with our metric.

Table 4: Example Inputs and Outputs for Always Same.

| Environment | Always Same |
|---|---|
| Input | $[x, x, x, y, y, y, z, z, z]$ |
| Temporal Referencing | $[m_1, m_4, m_4, m_2, m_4, m_4, m_3, m_4, m_4]$ |
| No Temporal Referencing | $[m_1, m_2, m_3, m_4, m_5, m_6, m_7, m_8]$ |

Table 5: Example Inputs and Outputs for Never Same.

| Environment | Never Same |
|---|---|
| Input | $[\boldsymbol{x}, \boldsymbol{y}, \boldsymbol{z}, \boldsymbol{a}, \boldsymbol{b}, \boldsymbol{c}, \boldsymbol{d}, \boldsymbol{e}]$ |
| Temporal Referencing | $[\boldsymbol{m}_1, \boldsymbol{m}_1, \boldsymbol{m}_1, \boldsymbol{m}_2, \boldsymbol{m}_2, \boldsymbol{m}_2, \boldsymbol{m}_3, \boldsymbol{m}_3, \boldsymbol{m}_3]$ |
| No Temporal Referencing | $[\boldsymbol{m}_1, \boldsymbol{m}_2, \boldsymbol{m}_3, \boldsymbol{m}_4, \boldsymbol{m}_5, \boldsymbol{m}_6, \boldsymbol{m}_7, \boldsymbol{m}_8]$ |

## C  ARCHITECTURE OVERVIEW

In Figure 5, we present an overview of our whole experimental setup. We can see the sender and receiver architectures, together with their inputs, as described in Section 2.4. We also show our loss calculations, for both the *Temporal-NL* version and the *Temporal* version of our games. In the *Temporal-NL* variant, we disable the temporal prediction module for the receiver, also disabling the temporal prediction loss. Consequently, in the *Non-Temporal* version we disable both the temporal prediction loss and the sequential LSTM, to achieve an architecture as close as possible to the ones used in most emergent communication research (Kharitonov et al., 2019).

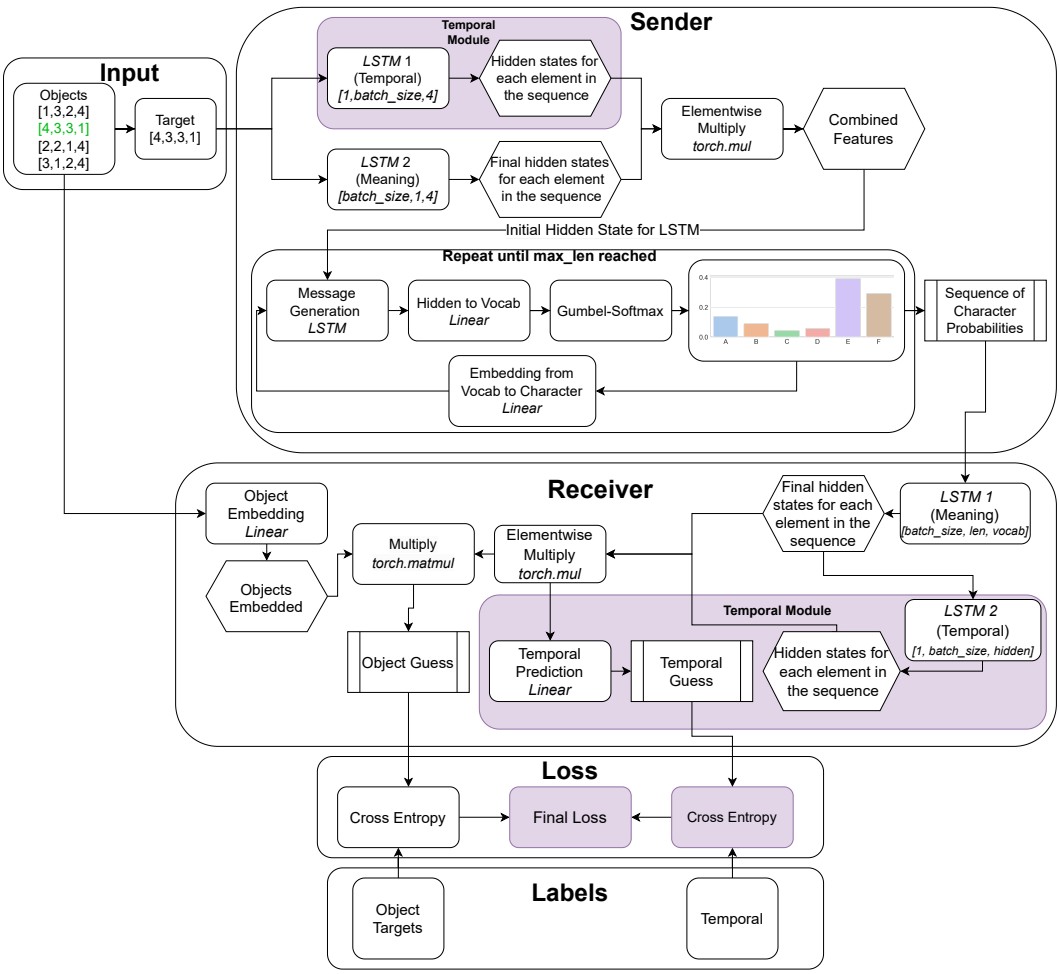

Figure 5: Full overview of our Temporal Referential Games setup. Together with the sender and receiver we have described in Section 2.4, we also include the working of the loss.

# D    ACCURACY ANALYSIS

We compare the accuracy of both variants of the *Temporal* networks to *Non-Temporal* networks in Figure 6. According to this metric, the *Temporal* networks with the temporal prediction loss perform marginally worse than the networks which do not include the temporal predictions. *Temporal-NL* networks that do not need to output temporal predictions, and so are not incentivised to assign more weight to the temporal aspects, perform better, matching the performance of the regular agents. Additionally, using the comparison between "RG Classic" and "RG Hard" (and analogously "TRG Previous" and "TRG Hard"), we observe that temporal references do not improve the performance on harder tasks, where targets are highly similar.

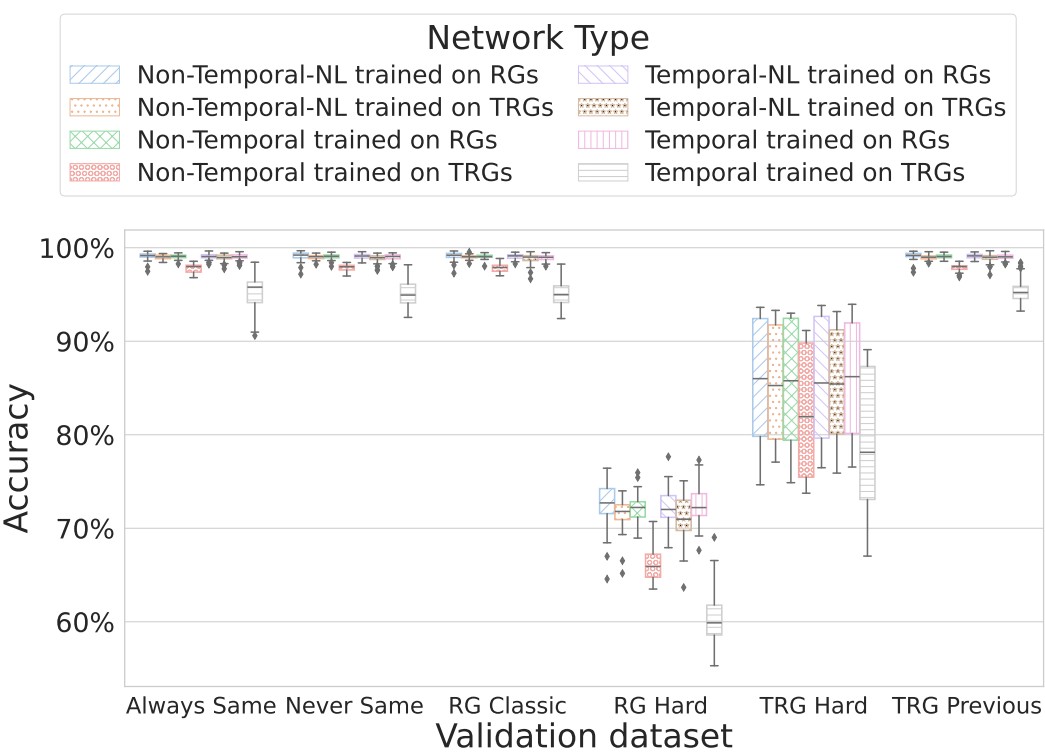

Figure 6: The evaluation accuracies for all of our network types, in all of our evaluation environments.

We believe that the reason for the accuracy drop lies in too much pressure on the temporal aspects in the case of networks that include a loss for temporal predictions. Because of this additional loss, the agents can increase their rewards by only focusing on creating temporal messages, without learning a general communication protocol. This then leads to an overfit to the training dataset, where they can rely on both their mostly temporal language and their memory of the object sequences, instead of communicating about the object attributes. Consequently, we observe a decline in performance on the evaluation dataset.

We also found no overfitting of our agents, even over 600 epochs of training, as their evaluation accuracy does not decrease. We find that our agents continue to hold at approximately 100% accuracy, even 500 epochs after they have reached peak performance, which we can see in Figure 7. This contrasts with a recent result in Rita et al. (2022), where agents were reported to overfit as training passed 250 epochs. As our *Temporal-NL* networks do not experience this decrease in accuracy, it may point to an advantage of temporal communication in countering co-adaptation of agents. However, the setting used in Rita et al. (2022) is different from ours, as the authors focus their analyses on a reconstruction game, where agents are tasked with reconstructing the sender input given a message. Instead, in our game, agents are asked to pick the correct object from a list. Another possible factor we have identified is the model size difference between our setting and Rita et al. (2022). While we

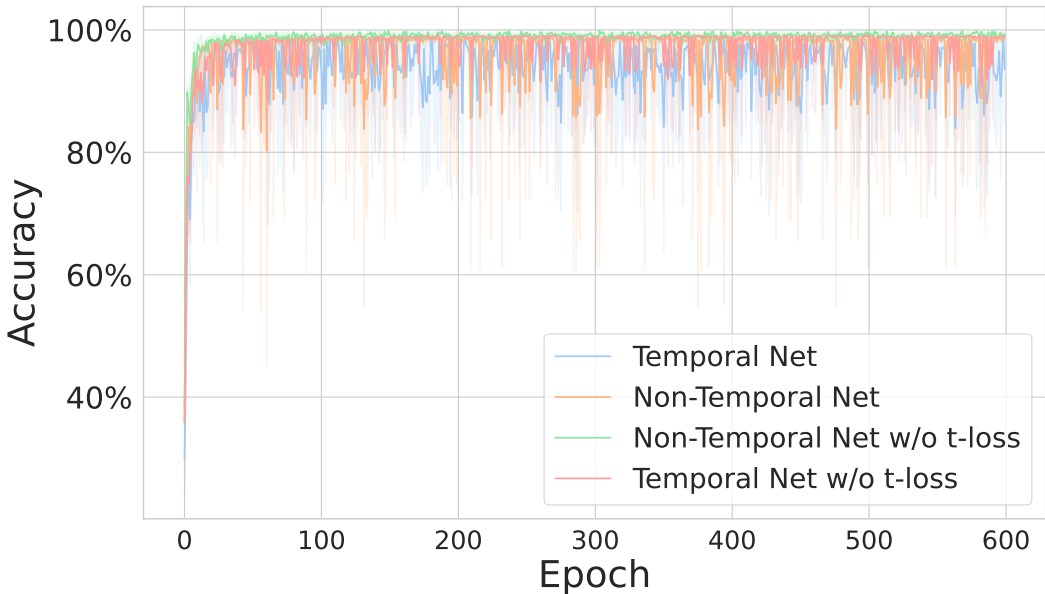

Figure 7: Average validation accuracy across all training and validation environments for all of our network types versus the number of epochs trained. All runs converge to close to 100% accuracy, and we observe no reduction in accuracy over longer training times.

use a hidden size of 128 for the LSTM, Rita et al. (2022) use 256. This is another possible reason for the observed overfitting, as it is well-known that larger models tend to overfit more easily. This may be in addition to the regularising impact of the environment or the temporal prediction loss.

# E    ANALYSIS FOR PREVIOUS HORIZON FROM $h_v = 1$ TO $h_v = 8$

In this section, we present the additional results for previous horizon from $h_v = 1$ to $h_v = 8$.

Table 6: Emergence of temporal references for a given horizon

| Network Type | $h_v = 1$ | $h_v = 2$ | $h_v = 3$ | $h_v = 4$ | $h_v = 5$ | $h_v = 6$ | $h_v = 7$ | $h_v = 8$ |
|---|---|---|---|---|---|---|---|---|
| Non-Temporal-NL | 0% | 0% | 0% | 0% | 0% | 0% | 0% | 0% |
| Non-Temporal | 0% | 0% | 0% | 0% | 0% | 0% | 0% | 0% |
| Temporal-NL | 99.44% | 100% | 100% | 99.72% | 98.61% | 100% | 99.72% | 98.89% |
| Temporal | 99.72% | 100% | 99.72% | 99.17% | 99.44% | 99.72% | 99.72% | 99.44% |

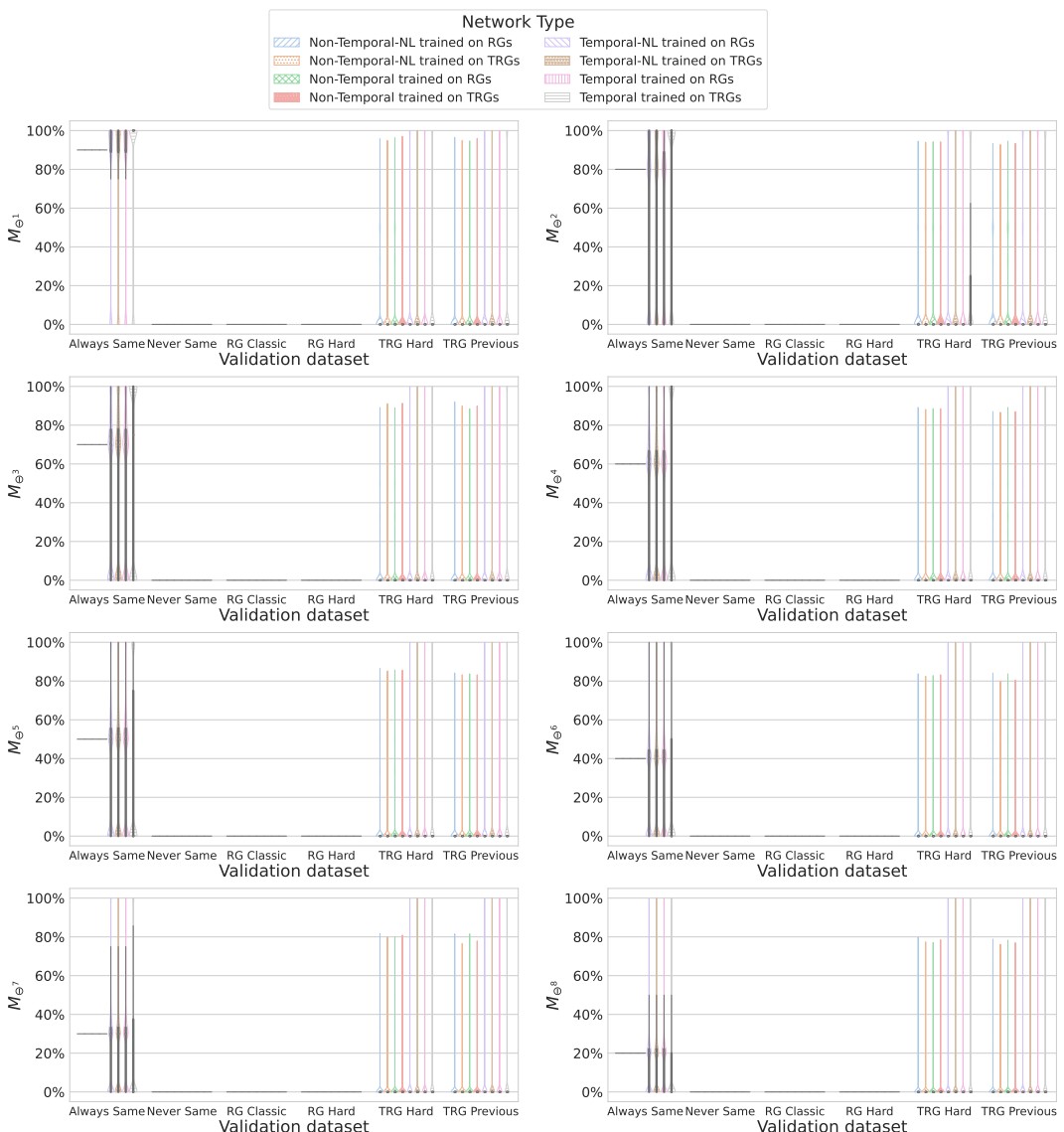

Figure 8: The $M_{\ominus^h}$ metric values per message, for all environments.

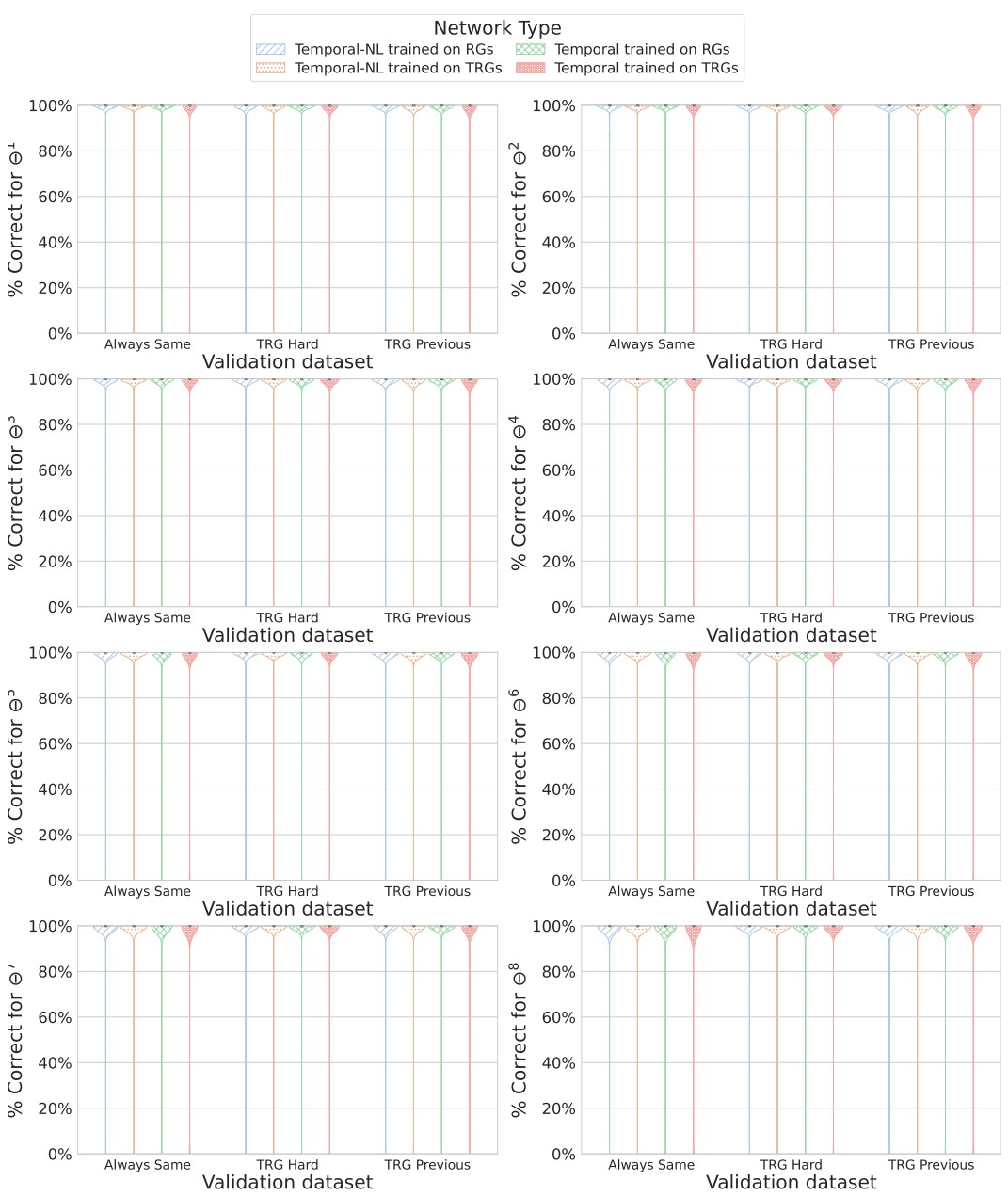

Figure 9: Correctness of messages used as the $\ominus^h$ operator.

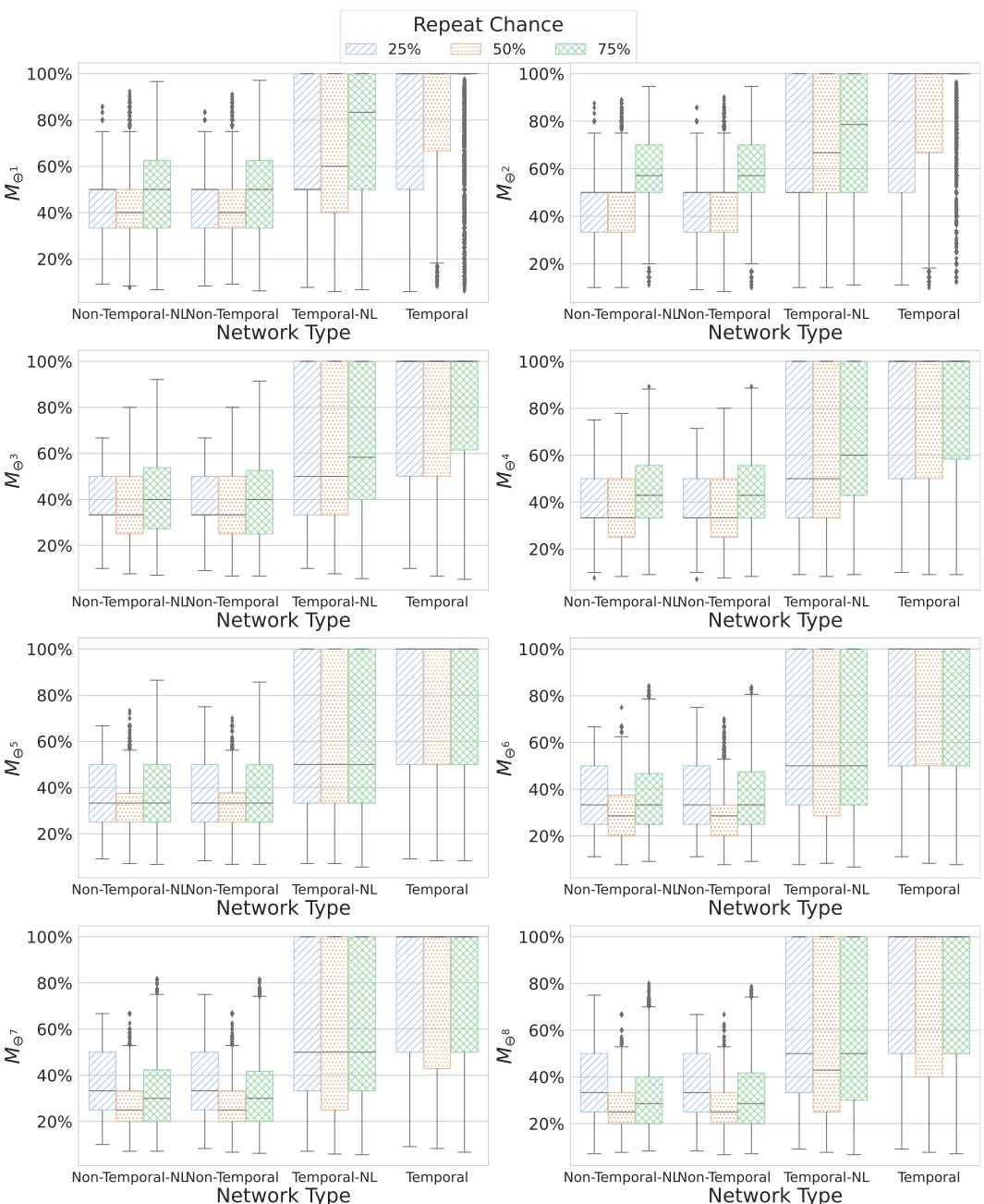

Figure 10: The $M_{\ominus^h}$ value when varying the network type and the chance of repetition.

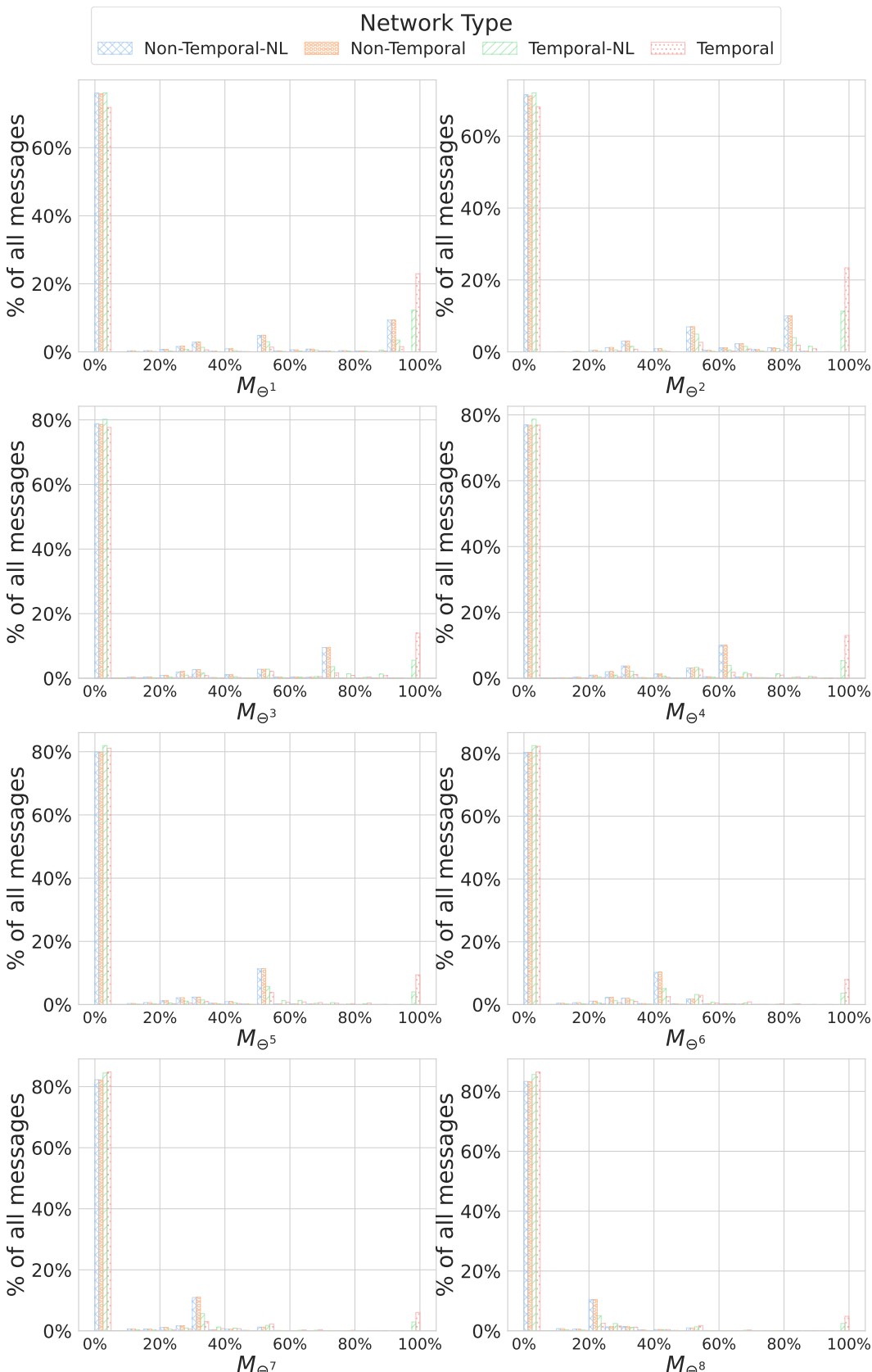

Figure 11: Usage of messages compared to their $M_{\ominus^h}$ value.

# F   COMPOSITIONALITY ANALYSIS

We analyse the created languages in terms of their compositionality scores, using the topographic similarity metric (Brighton and Kirby, 2006), commonly employed in emergent communication. We also use the *posdis* and *bosdis* metrics (Chaabouni et al., 2020), which account for languages where the symbols themselves carry all the information (permutation invariant languages — *bosdis*), or which use the positional information of individual characters (*posdis*). We show the results for these metrics in Figure 12, Figure 13, and Figure 14, respectively.

All of our agents create compositional languages with varying degrees of compositional structure, which shows that learning to use temporal references does not negatively impact this language property. Most agents reach values between 0.1 and 0.2 (the higher, the more compositional the language is) on the topographic similarity metric (Brighton and Kirby, 2006; Rita et al., 2022), where a score of 0.4 has been considered high in previous research (Rita et al., 2022). However, the topographic similarity metric would fail to measure compositionality regarding temporal references. As the temporal messages can be compositional but would not refer to a specific object, topographic similarity would not be able to identify them correctly. This could be the reason for the lower values compared to previous research (Rita et al., 2022).

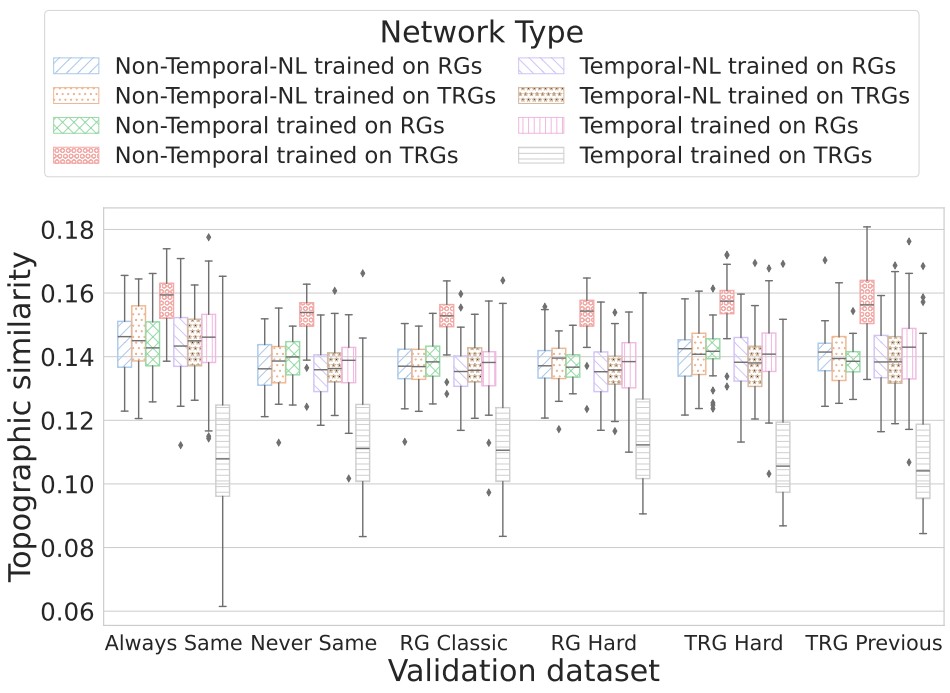

Figure 12: Topographic similarity scores for each network and evaluation environment.

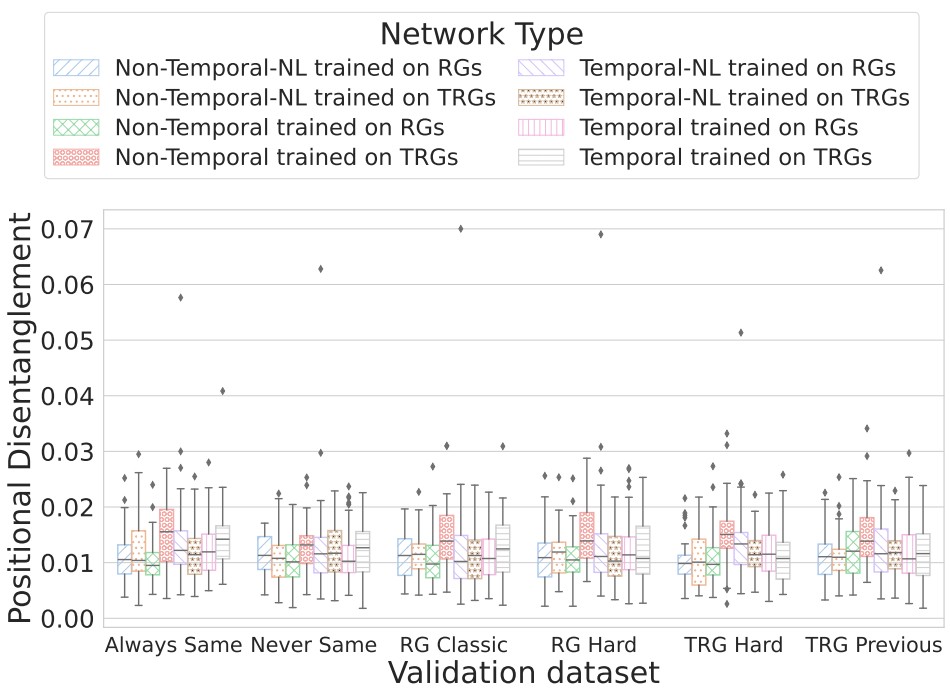

Figure 13: Positional disentanglement scores for each network and evaluation environment.

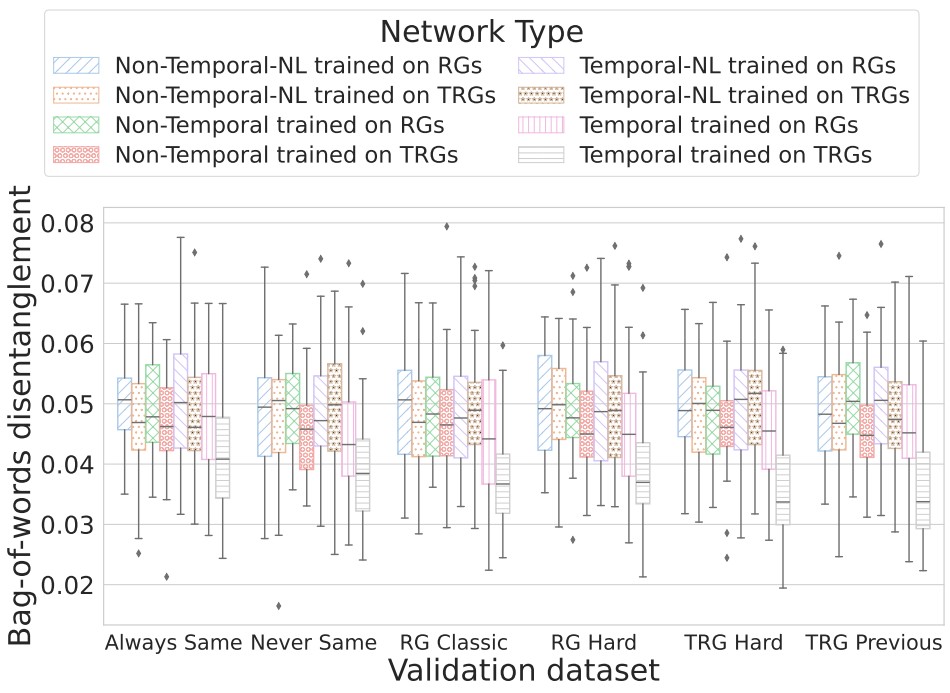

Figure 14: Bag-of-words disentanglement scores for each network and evaluation environment.

