# OpenReview forum: "It's About Time: Temporal References in Emergent Communication"
_ICLR.cc/2024/Conference — Submitted to ICLR 2024_

### Official Review · Reviewer_KopV · 2023-10-27

**Soundness:** 2 fair
**Presentation:** 3 good
**Contribution:** 3 good
**Rating:** 5
**Confidence:** 2

**Summary:**

This work considers the role of temporal references in emergent communication. For example, just as humans may say, "the thing I told you five minutes ago," EC agents should be able to refer to prior expressions as well.

To formalize this problem, the authors propose a metric, $M_{\theta^n}$ for measuring how often a message corresponds to a previous operator, where $n$ specifies how many timesteps previous.

The authors further propose to increase the use of temporal expressions through changes in architecture and loss. A "temporal module" in the sender and receivers model temporal processing, and a temporal loss can explicitly guide the receiver temporal module.

Overall, the authors find that neural architectures with the temporal modules use previous expressions far more often than baseline architectures.

**Strengths:**

## Originality
This work is pretty original. I am unaware of others considering the problem of temporal references in EC, and I find it an interesting area. Adding further RNN modules to the neural agents to represent temporal relations also seems clever (although I have important questions about the implementation, which I raise later).

## Quality
The results from the work are strong, per the desired metrics of temporal references.

## Clarity
The writing is largely clear, and the diagrams (both Fig 1 and 5, in the appendix) help explain the architectural changes. I think the authors did a very good job explaining simple aspects of LTL; I have some background in this area, and this seemed like a good presentation of necessary details.

## Significance
I quite like the combination of temporal logic methods with EC, and I think this is an interesting area for future research. The approach also seems simple (in a good way!), which should aid wider adoption of the ideas.

**Weaknesses:**

## Clarity around actual approach/implementation
My biggest concern with this work is about understanding the actual approach the authors used for the neural implementation. Fundamentally, I do not understand how the authors order inputs over time and the structure of communication. I explain my question about this under "Questions" - generally, I view this uncertainty as a weakness given how central such details are to the paper. Certainly, I could be misunderstanding information presented in the paper, but I am very knowledgeable about EC, so I do not think I am missing something obvious.

Because of my uncertainty about this crucial part of the paper, I am currently leaning against acceptance. I emphasize, however, that if the authors clarify their approach and it seems sound, I am willing to revise my score.

## Clarity around notation
The authors dedicate a fair amount of effort to establishing some notation based on LTL, but then do not seem to use it fully. For example, many of the figures (e.g., Figure 2) have almost natural-language y axis labels, whereas I would have expected a single mathematical symbol.

## Clarity of examples
The authors included a few simple examples in the paper that I first found very helpful, but now I think they actually are misleading. For example, in Section 3.1, the authors write about an intuitive example where the target sequence is [1, 2, 3, 2, 2, 1] and different message sequences are considered ([1, 2, 3, 2, 4, 4, 1], [1, 2, 3, 4, 4, 4, 1], etc.). My problem with this example is that, when I read it, it made me think that the speakers only generated a single token per target. This was reinforced in Table 4 in Appendix B.1, where the same type of example appears. However, in Section 3.3., where actual results are presented, I realized that a single message was composed of multiple tokens (5 or 6). That is fine, I wish I had realized this earlier on.

## Further metrics?
The authors mostly focus on the emergence of temporal references, but further metrics would help situate the broader network performance. Notably, the authors do evaluate validation set accuracy in Figure 6, and they show a small decrease in performance for temporal networks relative to non-temporal networks. This is good analysis but then makes me 1) worry about the utility of this approach, if accuracy goes down and 2) slightly contradicts some of the claims in the main paper, which motivate temporal references to improve performance.

## Minor:
1. Footnote 1 appears to indicate that the authors will share code upon acceptance. I suggest the authors use https://anonymous.4open.science/ in future submissions to link to anonymized code during the review phase.

2. The figures in Appendix E are not very clear and should be cleaned up for publication. I would also strongly caution about interpreting tSNE plots as indicative of compositionality; do the authors have references supporting this link?

3. The notation around equation 2 is not right. In particular, typically I read the right text in case-based equations as specifying the conditions under which that behavior is selected. I think the cases should just be $c < 0.5$ and $c >= 0.5$, therefore, without any of the $x$ parts.

4. Equation 3 seems to have a few things off about it. First, "objectRepeated" seems misleading as a name. I believe it just returns true if $x_n$ is present $h_v$ episodes ago, correct? What does that have to do with repetition? Second, the variable $n$ is very overloaded in the expression, which impedes readbility. It is simulatenously used to represent a fixed number of previous timestep (see left side of the equation) and is used as a counter variable (see right side of the equation).

**Questions:**

1. My main question, which I hope the authors address in their rebuttal, is about better understanding the exact form of the inputs and outputs of the speaker network. The key idea of this paper is that speakers should be able to refer back to previous communication if they see similar inputs. Thus, the speaker must somehow see a series of inputs, right? How is that represented? Is it a batch of inputs? I'm guessing that is the case, because the temporal module in the speaker reshapes the inputs based on batch size, so that makes me think it allows the temporal module to represent at which point in "time" an input appears. If it is the case, however, that the temporal order of inputs is represented by position in the batch, I am somewhat disappointed in the baselines used for comparison. It seems obvious non-temporal networks, which observe inputs in parallel over a batch, cannot represent temporal relations.

Overall, I am unsure about what the authors are actually doing to represent sequentially-ordered inputs, though, and really need clarity on this topic to evaluate this paper.

2. In the main paper, the authors typically present results for $h_v = 4$, but the authors mentioned that they conducted experiments for other. Results for such values are included in Appendix F, but the results end up raising lots of questions for me. Can the authors comment as to why results are seemingly so bimodal? Everything for $h_v <=4$ matches one profile, and everything for $h_v > 4$ matches another. Also, what was the actual training setup for these results?

---

> ### Author Response · Authors · 2023-11-14
> **Response to Reviewer KopV 1/3**
>
> We are thankful to the reviewer for their attention to detail and insightful feedback.
>
> We appreciate that the reviewer has found our direction to be novel and original. We also thank the reviewer for judging the clarity and significance to be of high quality.
>
> To clarify the points raised by the reviewer:
>
> ## Weaknesses
>
> ### Clarity around actual approach/implementation
>
> > My biggest concern with this work is about understanding the actual approach the authors used for the neural implementation. Fundamentally, I do not understand how the authors order inputs over time and the structure of communication. I explain my question about this under "Questions" - generally, I view this uncertainty as a weakness given how central such details are to the paper. Certainly, I could be misunderstanding information presented in the paper, but I am very knowledgeable about EC, so I do not think I am missing something obvious.
>
> We respond to this weakness in the questions section.
>
> > Because of my uncertainty about this crucial part of the paper, I am currently leaning against acceptance. I emphasize, however, that if the authors clarify their approach and it seems sound, I am willing to revise my score.
>
> We hope our responses to you, as well as the other reviewers, will help in clarifying any uncertainties.
>
> ### Clarity around notation
>
> > The authors dedicate a fair amount of effort to establishing some notation based on LTL, but then do not seem to use it fully. For example, many of the figures (e.g., Figure 2) have almost natural-language y axis labels, whereas I would have expected a single mathematical symbol.
>
> Thank you for this suggestion. We will convert our notation for the figures to use the LTL based notation, as well as in other points in the text.
>
> ### Clarity of examples
>
> > The authors included a few simple examples in the paper that I first found very helpful, but now I think they actually are misleading. For example, in Section 3.1, the authors write about an intuitive example where the target sequence is $[1, 2, 3, 2, 2, 1]$ and different message sequences are considered ($[1, 2, 3, 2, 4, 4, 1]$, $[1, 2, 3, 4, 4, 4, 1]$, etc.). My problem with this example is that, when I read it, it made me think that the speakers only generated a single token per target. This was reinforced in Table 4 in Appendix B.1, where the same type of example appears. However, in Section 3.3., where actual results are presented, I realized that a single message was composed of multiple tokens (5 or 6). That is fine, I wish I had realized this earlier on.
>
> We apologise for the confusion with our examples. We will improve this in the revised version, by using a different notation — as in our response to Reviewer `Avm5`. We will also include a thorough explanation of how the messages are composed. This will be available in the revised version.
>
> ### Further metrics?
>
> > The authors mostly focus on the emergence of temporal references, but further metrics would help situate the broader network performance. Notably, the authors do evaluate validation set accuracy in Figure 6, and they show a small decrease in performance for temporal networks relative to non-temporal networks. This is good analysis but then makes me 1) worry about the utility of this approach, if accuracy goes down and 2) slightly contradicts some of the claims in the main paper, which motivate temporal references to improve performance.
>
> While the accuracy does go down, we would consider this a negligible performance decrease, and only occurs when using the temporal prediction loss. This is especially true, considering that Temporal networks without the temporal prediction loss perform extremely similarly to our baselines, if not slightly better on the `RG Hard` dataset. Our claims about the increase in performance refer to improvements in bandwidth efficiency, were more complex environments considered, or if a length loss was applied. In these cases, temporal messages could 1) enable communication about previous events which would not be otherwise possible and 2) temporal references could be shorter when bandwidth efficiency is encouraged. In the revised version, we can further include metrics around compositionality, showing that temporal references do not impact it significantly (except for Temporal networks with the temporal prediction loss).

---

> ### Author Response · Authors · 2023-11-14
> **Response to Reviewer KopV 2/3**
>
> ### Minor:
>
> 1. > Footnote 1 appears to indicate that the authors will share code upon acceptance. I suggest the authors use <https://anonymous.4open.science/> in future submissions to link to anonymized code during the review phase.
>
>    We apologise for the confusion. The code is indeed hosted at <https://anonymous.4open.science/>, and the link in Footnote 1 is hyperlinked to our repository at https://anonymous.4open.science/r/TRG-E137.
>
> 2. > The figures in Appendix E are not very clear and should be cleaned up for publication. I would also strongly caution about interpreting tSNE plots as indicative of compositionality; do the authors have references supporting this link?
>
>    Thank you for pointing this out. We will clear up the figure in Appendix E, and remove the claim of the clustering being an indication of compositionality.
>
> 3. > The notation around equation 2 is not right. In particular, typically I read the right text in case-based equations as specifying the conditions under which that behavior is selected. I think the cases should just be $c < 0.5$ and  $c >= 0.5$, therefore, without any of the $x$ parts.
>
>    We will remove the $x$ parts from the right side of the equation in the revised version.
>
> 3. > Equation 3 seems to have a few things off about it. First, "objectRepeated" seems misleading as a name. I believe it just returns true if $x_n$ is present $h_v$ episodes ago, correct? What does that have to do with repetition? Second, the variable $n$ is very overloaded in the expression, which impedes readbility. It is simulatenously used to represent a fixed number of previous timestep (see left side of the equation) and is used as a counter variable (see right side of the equation).
>
>    We will rename the `objectRepeated` function to `objectSame` as that is what it would be checking, i.e., if the object $x_n$ is the same as the object $x_{n-h_v}$. We will also split the variable $n$ to improve readability.

---

> ### Author Response · Authors · 2023-11-14
> **Response to Reviewer KopV 3/3**
>
> ## Questions
>
> 1. > My main question, which I hope the authors address in their rebuttal, is about better understanding the exact form of the inputs and outputs of the speaker network. The key idea of this paper is that speakers should be able to refer back to previous communication if they see similar inputs. Thus, the speaker must somehow see a series of inputs, right? How is that represented? Is it a batch of inputs? I'm guessing that is the case, because the temporal module in the speaker reshapes the inputs based on batch size, so that makes me think it allows the temporal module to represent at which point in "time" an input appears. If it is the case, however, that the temporal order of inputs is represented by position in the batch, I am somewhat disappointed in the baselines used for comparison. It seems obvious non-temporal networks, which observe inputs in parallel over a batch, cannot represent temporal relations.
>
>    Your description of the way we represent the series of inputs is correct. We indeed represent the temporal order of inputs by their position in the batch.
>
>    However, we would disagree with this being disappointing. We agree that non-temporal networks cannot represent temporal relations. However, we include them as both a baseline in terms of the temporal metric, and **to prove** this intuition. We only include this simple trick, instead of more complex ways to learn temporal relationships (such as using attention networks, single LSTMs, or transformers, which we touch on in our response to Reviewer `RdRa`), to show that with just a simple modification to the LSTM the agents learn to exploit temporal relationships. We also show that no other losses are necessary. We consider the simplicity of both our setup and the ease of emergence of temporal relationships a positive aspect of our work, showing that the learnings are easily transferable to more complex settings and agent architectures.
>
> > Overall, I am unsure about what the authors are actually doing to represent sequentially-ordered inputs, though, and really need clarity on this topic to evaluate this paper.
>
> Please refer to our response above.
>
> 2. > In the main paper, the authors typically present results for $h_v = 4$, but the authors mentioned that they conducted experiments for other. Results for such values are included in Appendix F, but the results end up raising lots of questions for me. Can the authors comment as to why results are seemingly so bimodal? Everything for $h_v <= 4$ matches one profile, and everything for $h_v > 4$ matches another. Also, what was the actual training setup for these results?
>
>    The results initially appeared bimodal, due to an issue in the analysis. We miscounted some runs, which only had $h_v$ up to $4$ as also counting for runs which had $h_v$ up to $8$. This meant that the values for $h_v$ between $5$ and $8$ were only half of what they should have been. We have now rectified this, and performed additional runs, for 10 runs per configuration. We present the corrected and updated results in the table below. The results no longer look bimodal.
>
>    In terms of the training setup — we have run each agent configuration with $h_v=8$. We will also update our Grid Search Parameters in `Appendix A, Table 2`, with this information.
>
>
> | Network Type    | $h_v=1$ | $h_v=2$ | $h_v=3$ | $h_v=4$ | $h_v=5$ | $h_v=6$ | $h_v=7$ | $h_v=8$ |
> |-----------------|---------|---------|---------|---------|---------|---------|---------|---------|
> | Non-Temporal-NL | 0%      | 0%      | 0%      | 0%      | 0%      | 0%      | 0%      | 0%      |
> | Non-Temporal    | 0%      | 0%      | 0%      | 0%      | 0%      | 0%      | 0%      | 0%      |
> | Temporal-NL     | 99.44%  | 100%    | 100%    | 99.72%  | 98.61%  | 100%    | 99.72%  | 98.89%  |
> | Temporal        | 99.72%  | 100%    | 99.72%  | 99.17%  | 99.44%  | 99.72%  | 99.72%  | 99.44%  |

---

> > ### Comment · Reviewer_KopV · 2023-11-15
> >
> > Thanks for including a link to the code. Browsing through quickly, the code is impressively clean and well-commented.
> >
> > Thanks for the clarifications and the new results for different $h_v$ values.
> >
> > The response about batches and temporal ordering clarifies my uncertainty.
> >
> > Given that the non-temporal networks have literally no way of accessing temporal data, I still find the comparisons to baselines less important than it seems in the paper. Perhaps the authors are already getting at this in the rebuttal, but I might emphasize the stand-alone positive nature of their results rather than saying it's an improvement over baselines. To pick just one example, whereas the authors write in the abstract "a different agent architecture is sufficient for the natural emergence of temporal references..." I might suggest they say something like "when agents receive time-extended inputs, standard agent architectures with no additional losses are sufficient to induce temporal references." Obviously, the authors should choose their own words. I am simply trying to emphasize that, to me, the contribution is not so much about a novel architecture but rather just the problem formulation and inclusion of temporal data as an important input.
> >
> > Given the clarifications by the authors, I have increased my score from a 3 to a 5. I believe I now understand the authors' approach, which seems technically correct but somewhat limited in scope. To increase my score, I would want to see further changes that are beyond the scope of a rebuttal, such as 1) further framing or metrics quantifying actual benefits of temporal references, 2) exploration of different time-compatible architectures, and 3) comparison to many existing EC works in time-extended MARL settings beyond reference games in which agents could likely exploit temporal references as well.

---

### Official Review · Reviewer_WtjD · 2023-10-28

**Soundness:** 2 fair
**Presentation:** 2 fair
**Contribution:** 3 good
**Rating:** 3
**Confidence:** 4

**Summary:**

The current study is the first investigation of the temporal reference in emergent communication, in which its contributions are three-fold:
- it formally defines an environment for emergent communications in referential grams that consider referent in the past using Linear Temporal Logic.
- it proposes a new architecture for the agent
- it conducted a temporality analysis of the agent

**Strengths:**

- It is the first work to investigate the temporal references in emergent communication, during which it defines an environment and introduces an architecture of the agent. The subject matter is interesting and vital.
- Despite the limitations listed below, the experiments to a certain extent provide a response to the major research question of this study and support for the conclusions.

**Weaknesses:**

1. The notations in this paper need to be further standardised and unified. The current ones sometimes make readers (at least me) hard to follow. Here are the ones that I have spotted: (1) a vector in this paper sometimes means the feature vector of a target and sometimes a target sequence. Both of them can have arbitrary lengths and often appear without further explanation. (2) things like x \in V are not conditions in equation 2, IMO, they should not connected with the "real" conditions with AND. Also, see my questions below.
2. The paper spends quite a lot of space describing the details of the agents (e.g., the dimension of each vector during the implementation). Contrary to this is the absence of motivations for many choices made in the current study. For example, why the environment was defined in this way? Why a second LSTM in both sender and receiver? Why does there have to be an additional LSTM to capture temporal information? Actually, in terms of the architectural design, the two additional LSTMs in the sender and receiver have very different roles as the two LSTMs in the sender are parallel, and the two in the receiver are stacked.
3. Related to my second concern, the experiment design in the present study also cannot validate the rationalities of these choices. This makes the conclusions from the experiments, namely a different batching strategy is sufficient, look narrow.

### Typos:
- Page 3: c \leq 0.5 -> c < 0.5
- Page 6: When you are saying Temporal Loss, I believe you meant temporal prediction loss rather than temporal loss.

**Questions:**

- I do not fully understand the definition of feature space in section 2.1. If F is a feature space of a feature, then why its size N_F is the number of features rather than the number of values that the associated property can take?

---

> ### Author Response · Authors · 2023-11-14
> **Response to Reviewer WtjD**
>
> We would like to thank the reviewer for their insightful comments and valuable criticisms that will enhance the quality of our work.
>
> We also appreciate that the reviewer has found our work novel, and that the study of temporal references is interesting and vital. We are also thankful that our experiments were judged to provide a partial answer to a major research question.
>
> To clarify the points raised by the reviewer:
>
> ## Weaknesses
>
> 1. > The notations in this paper need to be further standardised and unified. The current ones sometimes make readers (at least me) hard to follow. Here are the ones that I have spotted: (1) a vector in this paper sometimes means the feature vector of a target and sometimes a target sequence. Both of them can have arbitrary lengths and often appear without further explanation. (2) things like x \in V are not conditions in equation 2, IMO, they should not connected with the "real" conditions with AND. Also, see my questions below.
>
>    Thank you for the suggestion. We will unify the notation across the paper, and remove the extraneous information in Equation 2 in the revised version.
>
> 2. > The paper spends quite a lot of space describing the details of the agents (e.g., the dimension of each vector during the implementation). Contrary to this is the absence of motivations for many choices made in the current study. For example, why the environment was defined in this way? Why a second LSTM in both sender and receiver? Why does there have to be an additional LSTM to capture temporal information? Actually, in terms of the architectural design, the two additional LSTMs in the sender and receiver have very different roles as the two LSTMs in the sender are parallel, and the two in the receiver are stacked.
>
>    We define our `TRG Previous` environment to allow agents to develop temporal references. We additionally use notation from PLTL to bring rigour to our definitions.
>    In terms of the LSTMs — the second, temporal, LSTM is needed to capture temporal information as it is batched differently to the regularly employed LSTM, which in our case is the meaning LSTM. There does not necessarily need to be an additional LSTM, but it does improve performance, with one LSTM focusing on the meaning in the objects, and the other on their temporal relationships. We explore this briefly in `Section 3.3, p7`.
>
>     Indeed, the two LSTMs perform very different roles, as the sender agent has to understand the temporal relationships and the meaning of the object separately using two parallel LSTMs. The receiver LSTM first decodes the messages and then observes the temporal relationships between them. This was done to stay as close as possible to our base agents, as presented in EGG [1].
>
> 3. > Related to my second concern, the experiment design in the present study also cannot validate the rationalities of these choices. This makes the conclusions from the experiments, namely a different batching strategy is sufficient, look narrow.
>
>    We disagree that the experiment design makes our conclusions narrow. As touched on in our response to Reviewer `KopV`, we present this simple architecture to show that no significant changes are needed for the temporal references to emerge. While we have tested other architecture, such as transformer or attention-based, the simplicity of our model and heuristics allows for an easy transfer of these learnings to other settings.
>
> ## Questions & Typos
>
> > I do not fully understand the definition of feature space in section 2.1. If F is a feature space of a feature, then why its size N\_F is the number of features rather than the number of values that the associated property can take?
>
> We apologise for the confusion. We will clarify our definitions of features and properties by moving to the more standard attribute-value notation. In this case, the feature space would translate to the values that each attribute can have.
>
> ### Typos:
>
> * > Page 3: c \leq 0.5 -> c < 0.5
> * > Page 6: When you are saying Temporal Loss, I believe you meant temporal prediction loss rather than temporal loss.
>
> Thank you for pointing out the typos. We will correct them in the revised version.
>
> [1] Kharitonov et al., EGG: a toolkit for research on Emergence of lanGuage in Games,  EMNLP-IJCNLP 2019

---

> > ### Comment · Reviewer_WtjD · 2023-11-16
> > **Reponse to the rebuttal**
> >
> > Thanks for your response.
> >
> > > We disagree that the experiment design makes our conclusions narrow.
> >
> > Maybe there is a misunderstanding. I meant the experimental design cannot conclude that the current model design is rational (e.g., the choice of the stacked LSTM and parallel LSTM). I am asking for a more advanced architecture.

---

### Official Review · Reviewer_RdRa · 2023-10-28

**Soundness:** 3 good
**Presentation:** 3 good
**Contribution:** 2 fair
**Rating:** 5
**Confidence:** 4

**Summary:**

This paper introduces a variant of the common referential game in emergent
communication game: the Temporal Referential Game (TRG).  The TRG is an
iterated version of the standard referential game where recent objects have
a higher probability of being seen again compared to non-recently seen objects.
The authors introduce a temporal LSTM module and a temporal loss to better
solve this game and observe the emergence of temporally referent messages.

**Strengths:**

- (major) Temporal reference is a pertinent feature of language, and therefore,
  it is good to use an environment with multiple timesteps with additional
  temporal structure (i.e., recently seen object are more likely to appear
  again).
- (minor) The Temporal Referential Game (TRG) is an appropriate extension to
  the regular referential game.  It minimally adds a temporal aspect to the
  game without changing it too much, making it good for pioneering basic
  concepts about temporal reference.

**Weaknesses:**

This paper does not make a substantial contribution to the field.  While the
TRG could elicit potentially interesting behaviors, the primary thrust of this
paper is that it introduces a new architecture which is able to solve this game
without much further analysis about temporality.  The problem here is that the
new architecture is, to the best of my knowledge, just an LSTM where each
timestep is a single round of the referential game.  This, to me, seems like an
appropriate baseline with which to work with and does not seem like
a contribution in and of itself.  Additionally, the temporal loss seems to work
against the claim that temporal reference "emerges from functional pressures";
this claim I see as being implicit in most all emergent communication research,
otherwise we are more or less looking at a trivial supervised learning problem.

**Questions:**

What is the significance of the temporal loss and the temporal LSTM?  What do
they offer beyond being a commonsense baseline for the TRG?

### Misc comments

- Sec 1
    - social deduction game: cite related work (e.g., [this](https://www.semanticscholar.org/paper/RLupus%3A-Cooperation-through-emergent-communication-Brandizzi-Grossi/d94504e5319ea1a2dc7df8a1ad25a48a4c3ae650))
- Sec 2.1
    - "attribute-value" seems more standard than "property-feature"; the latter
      sounds like "property" and "feature" could refer to the same thing
- Sec 2.3
    - Seems like you could just parameterize the threshold for $c$?
    - Alternatively, it would probably be even clearer to say that we sample
      $c$ from a Bernoulli distribution since that is in effect what is
      happening.  Also, saying $c$ is uniformly sampled integer from $[0,1]$ is
      sort of confusing since interval notation refers to real values, not
      integers.
    - "verify whether the messages" -> "determine whether or not the messages"
- Sec 2.4
    - "same as the EGG framework" is not a good way to specify the architecture
      since it is a whole framework and not a particular model.  Maybe just
      take a sentence or two to specify the defaults that are being used.
  p6
- Sec 3.1
    - The $100\\%$ seems too restrictive to determine a word is properly
      temporal reference.  For example, in English, is "yesterday" temporally
      referent?  Because it can be used in a non-temporally referent way
      (e.g., "TensorFlow is yesterday's ML library.").
- Sec 3.2
    - It seems like the non-temporal network is strictly precluded from
      performing temporal reference?  How is this a useful baseline, then?  Are
      we seeing if it can perform just as well or what the random baseline of
      temporal reference might be?
- Sec 3.3
  - Is $M_{\ominus^4}$ the max value of any message?  It seems like the
    temporality is per-message, not per-language.
- Figures 2a and 3a are difficult to read.

---

> ### Author Response · Authors · 2023-11-14
> **Response to Reviewer RdRa 1/2**
>
> We are thankful to the reviewer for their valuable insights and helpful critiques.
>
> We appreciate that the reviewer has found or work novel and a good extension to the current literature. We also appreciate that our experimental setup was found to be a good testbed for exploring temporal references.
>
> To clarify the points raised by the reviewer:
>
> ## Weaknesses
>
> > This paper does not make a substantial contribution to the field. While the TRG could elicit potentially interesting behaviors, the primary thrust of this paper is that it introduces a new architecture which is able to solve this game without much further analysis about temporality. The problem here is that the new architecture is, to the best of my knowledge, just an LSTM where each timestep is a single round of the referential game. This, to me, seems like an appropriate baseline with which to work with and does not seem like a contribution in and of itself. Additionally, the temporal loss seems to work against the claim that temporal reference "emerges from functional pressures"; this claim I see as being implicit in most all emergent communication research, otherwise we are more or less looking at a trivial supervised learning problem.
>
> We apologise for the confusion. Our focus in this paper was not to introduce a new architecture, but rather show how the emergence of temporal references could occur. While the architecture we present is indeed just an LSTM, we consider it a good baseline because it is a simple architectural change, and works to show that temporal references do not require new complex architectures. In addition, we evaluated architectures which contain _only_ the differently batched LSTM (`Section 3.3, p7`) which shows that allowing the agents to learn temporal references is the only required factor. Additionally, while we could not include this in the full text, we also considered different agent models, including transformers and attention-based architectures. While transformers would be very inefficient (workarounds have to be used to exploit the batching trick, as passing in a sequence of `batch_size` objects would generate a single `batch_size` message), attention-based networks work just as well as the simple LSTM.
>
> In terms of the temporal loss, we introduce it to verify that additional losses are not required, by showing that `Non-Temporal-NL` networks can also develop temporal references. We modify the loss function to show the behaviour is not tied to a specific one, and will occur with just the regular loss function. We consider our exploration of the possible impact of the loss function similar to that of [1-4].

---

> ### Author Response · Authors · 2023-11-14
> **Response to Reviewer RdRa 2/2**
>
> ## Questions & Misc Comments
>
> > What is the significance of the temporal loss and the temporal LSTM? What do they offer beyond being a commonsense baseline for the TRG?
>
> We explain why we use both in our response to the weaknesses.
>
> * Sec 1
>     * > social deduction game: cite related work`
>     * Thank you for pointing this out, we have added the citations.
>
> * Sec 2.1
>     * > "attribute-value" seems more standard than "property-feature"; the latter sounds like "property" and "feature" could refer to the same thing`
>     * We have changed our wording to be more standard by using "attribute-value" terminology.
>
> * Sec 2.3
>     * > Seems like you could just parameterize the threshold for $c$?`
>     * > Alternatively, it would probably be even clearer to say that we sample $c$ from a Bernoulli distribution since that is in effect what is happening. Also, saying is uniformly sampled integer from $[0,1]$ is sort of confusing since interval notation refers to real values, not integers.`
>     * Thank you for this suggestion. We will reword the description to use the Bernoulli distribution with $p=0.5$ for the base case and $p=0.25$ or $p=0.75$ for `Figure 3b`.
>     * > "verify whether the messages" -> "determine whether or not the messages"`
>     * We will change this wording in the revised version.
>
> * Sec 2.4
>
>     * > "same as the EGG framework" is not a good way to specify the architecture since it is a whole framework and not a particular model. Maybe just take a sentence or two to specify the defaults that are being used. p6
>     * Thank you for this suggestion. We will include a short description in the revised version.
>
> * Sec 3.1
>
>     * > The $100\\%$ seems too restrictive to determine a word is properly temporal reference. For example, in English, is "yesterday" temporally referent? Because it can be used in a non-temporally referent way (e.g., "TensorFlow is yesterday's ML library.").
>     * We agree that $100\\%$ is restrictive. However, as we mention in our response to Reviewer $Avm5$, we use $100\\%$ as the cut-off to be able to conclude with certainty that we do observe temporal references. We agree with the example, that in English some words that are usually temporal references can be used in non-temporal contexts. However, this would be challenging to separate from accidental temporal references using the $M_{\ominus^n}$  metric (as we discuss in `Section 4, p2`). Instead, we look for words which would represent, for example, `2 minutes ago`, which would not be used in a non-temporal context.
>
> * Sec 3.2
>
>     * > It seems like the non-temporal network is strictly precluded from performing temporal reference? How is this a useful baseline, then? Are we seeing if it can perform just as well or what the random baseline of temporal reference might be?
>     * As we touch on in response to Reviewer `KopV`, we include them as both a baseline in terms of the temporal metric, and to prove the intuition that they will not be able to form temporal references. Through this, we can show that only a simple modification is needed to the base agent to allow them to develop temporal references.
>
> * Sec 3.3
>
>     * > Is $M_{\ominus^4}$ the max value of any message? It seems like the temporality is per-message, not per-language.
>     * The $M_{\ominus^n}$ indeed calculates temporality per message. We use the metric to calculate the value for each message in a given language, and these are the values we present in the figures. This is as we do not aim to find the average value for $M_{\ominus^n}$ for a language, as that could be misleading, due to the reasons outlined in `Section 4, p2`. Instead, we look for messages which achieve $100\\%$ on the $M_{\ominus^n}$ metric.
>
> * > Figures 2a and 3a are difficult to read.
> * We will improve the legibility of the figures in the revised version.
>
>
>
>
>
> [1] E. Cheng, M. Rita, and T. Poibeau, ‘On the Correspondence between Compositionality and Imitation in Emergent Neural Communication’. ACL 2023
>
> [2] M. Tucker, R. P. Levy, J. Shah, and N. Zaslavsky, ‘Trading off Utility, Informativeness, and Complexity in Emergent Communication’, NeurIPS 2022
>
> [3] M. Rita et al., ‘Emergent Communication: Generalization and Overfitting in Lewis Games’, NeurIPS 2022
>
> [4] S. Vanneste et al., ‘Learning to Communicate Using Counterfactual Reasoning’, Adaptive and Learning Agents Workshop (ALA 2022)

---

> ### Comment · Reviewer_RdRa · 2023-11-16
> **Reading the paper in a different light**
>
> I have read both responses in full.
>
> > We apologise for the confusion. Our focus in this paper was not to introduce
> > a new architecture, but rather show how the emergence of temporal references
> > could occur.
>
> This was the contribution I had in mind in writing the review.  Generally
> speaking, I am of the mind that "possibility" demonstrations are not
> significant contributions to the field of emergent communication in 2023.  By
> "possibility" demonstration, I mean environment and experiments which are
> primarily showing that is possible for successful communication protocols to be
> developed in an incrementally developed environment.
>
>
> Insofar as this is the framing my evaluation does not change.  If I reconsider
> the paper as primarily being about the nature of the _particular temporal
> operator_ "previously", I do believe there is more merit in the contribution.
> I distinguish this from the framing the paper largely gives by more often
> saying that "temporal reference can emerge" with the "previously" operator more
> being one convenient way to define temporal reference.  The former has to do
> with a specific type of communication that has potential nuances that have not
> yet been explored while the latter almost goes without saying (i.e., of course
> RNNs will pick up on correlations across timesteps---this is their whole
> motivation for language modeling).
>
> Nevertheless, I find myself questioning how successfully this paper achieves
> the reframed goal mentioned above as it took multiple readings of the paper
> before I came across this "alternative" interpretation.  Furthermore, the
> empirical analysis seems as much geared towards supporting the statement
> "temporal reference emerges in some capacity" as it is toward investigating the
> nature of the "previously operator.  In light of this reframing, I will change
> my scores as follows (viz. the reframing is less clear but more significant):
>
> - Presentation: 4 -> 3
> - Contribution: 1 -> 2
> - Rating: 3 -> 5
> - Confidence: 5 -> 4

---

### Official Review · Reviewer_Avm5 · 2023-10-29

**Soundness:** 2 fair
**Presentation:** 2 fair
**Contribution:** 3 good
**Rating:** 6
**Confidence:** 3

**Summary:**

This paper introduces a novel dimension of emergent communication, focusing on temporal reference. In this framework, the sender is tasked with encoding temporal information about target objects, enabling the receiver to distinguish them from other distractors. To facilitate this, a temporal LSTM component is incorporated into the agents' architecture for temporal encoding. Furthermore, a loss function for temporal object classification is introduced to encourage the emergence of temporal encoding. To assess the effectiveness of this approach, an evaluation metric is devised to measure the frequency of using the same message to describe previously presented objects. The experimental findings highlight the role of the temporal LSTM in facilitating the development of temporal communication.

**Strengths:**

1. This paper introduces temporal reference into the emergent communication field, which is an important topic worth exploring. It leverages the sequential position of objects within the batch as a means of encoding temporal information, thereby modifying traditional referential games and their associated training frameworks.

2. The paper demonstrates a comprehensive design of the experiments. It encompasses various modifications to modeling architectures, incorporates multiple environments, and also analyzes different statistical results.

**Weaknesses:**

1. Example demonstration: the authors present multiple examples using integer arrays in the game design (section 2.3), agent architecture (figure 1), how to compute the temporality metric (section 3.1), as well as the sent messages (section 3.3), which I believe are of different meanings. However, using integer arrays in all the examples can make the demonstration more confusing. Different symbols used in different cases would be better. For example, objects as names, and messages using alphabets.

2. Experimental analysis clarification:

    a. In section 3.3, it is said “only networks that have the sequential LSTM are capable of producing temporal references”: From Figure 2a, we can observe all networks have cases “reaching % used as previous” above 80%. A clarification on how you make a judgment on “producing temporal references” should be made.

    b. In Figure 3b, only the results of Temporal networks are presented because “non-temporal networks learn no temporally specialised messages”. The threshold and significance between temporally and no temporally specialised messages should be made clear as well.

    c. Through Figure 3a, the claim that “these messages could be a more efficient way of describing objects” needs to be further clarified.

    d. In Figure 2, which of the six environments used for the legends about “trained RGs” and “TRGs” are not specified.

    e. During the experiment analysis, the results of “Never same” are not presented. Also, there is not much difference, or the difference is not underscored between “environment” and “environment Hard”. This part could be better organized based on the analysis needed.

**Questions:**

1. For the temporal loss, I assume it is a categorical classification about the last position the object shows up. I think it could be misleading when trained with target object classification together. There are some circumstances when the objects are the same but the temporal labels are different, which may cause the network to be confused. Have you tried to add a smaller weight to the temporal loss or try to add this loss on top of the temporal LSTM?

2. More analysis of the features that the temporal LSTM provides would be helpful to demonstrate its vital role.

---

> ### Author Response · Authors · 2023-11-14
> **Response to Reviewer Avm5 1/2**
>
> We would like to thank the reviewer for their insightful comments and constructive criticism.
>
> We appreciate that the reviewer found our work novel and that our topic is worth exploring. We also appreciate that the experiments and the method were seen to be comprehensive and a good demonstration of the concepts presented.
>
> To clarify the points raised by the reviewer:
>
> ## Weaknesses
>
> 1. > Example demonstration: the authors present multiple examples using integer arrays in the game design (section 2.3), agent architecture (figure 1), how to compute the temporality metric (section 3.1), as well as the sent messages (section 3.3), which I believe are of different meanings. However, using integer arrays in all the examples can make the demonstration more confusing. Different symbols used in different cases would be better. For example, objects as names, and messages using alphabets.
>
>     Thank you for this suggestion. We will revise our examples to use different symbols for the different cases.
>
> 2. Experimental analysis clarification:
>
>     > a. In section 3.3, it is said “only networks that have the sequential LSTM are capable of producing temporal references”: From Figure 2a, we can observe all networks have cases “reaching % used as previous” above 80%. A clarification on how you make a judgment on “producing temporal references” should be made.
>
>     Our cut-off for classifying a network as producing temporal references is $100\\%$, as a strong requirement, to ensure our reported results are significant. We choose to use the $100\\%$ to ensure that our positive observation is not a result of chance, such as in the case of the $80\\%$, where the temporal messages are misclassified due to naturally occurring repetitions in the dataset. We further explore this in `Section 4, p2`. We will clarify this in the revised version.
>
>     > b. In Figure 3b, only the results of Temporal networks are presented because “non-temporal networks learn no temporally specialised messages”. The threshold and significance between temporally and no temporally specialised messages should be made clear as well.
>
>     We present the Non-Temporal variants in `Figure 3b`, so we assume this refers to `Figure 2b`.
>
>     The threshold for `Figure 2b` is the same as we described in our response above - $100\\%$. Therefore, as the Non-Temporal networks never reach this threshold (exactly $0\\%$ of the networks) they are not included in this Figure. We will clarify this in revised version.
>
>     Could you please clarify what you mean by `significance between temporally and no temporally specialised messages`?
>
>     > c. Through Figure 3a, the claim that “these messages could be a more efficient way of describing objects” needs to be further clarified.
>
>     Since only a few messages are needed for the temporal references, they could be used more frequently. This message specialisation, combined with a linguistic parsimony pressure, could lead to a more efficient way of describing an object. For example, describing the object properties $[red, stripes, circle]$ requires more bandwidth than sending only the timestep the object last appeared $[t-8]$, assuming the agents are encouraged to create shorter messages due to an external pressure. We will clarify this in the revised version.
>
>     > d. In Figure 2, which of the six environments used for the legends about “trained RGs” and “TRGs” are not specified.`
>
>     Where we refer to the training environments `RGs` and `TRGs` in `Figure 2` we use the base environment, i.e., `RG Classic` and `TRG Previous`. The other variants (i.e., `RG Hard`, `TRG Hard`, `Never Same`, `Always Same`) are only used for evaluation, and not training. We will clarify this in the revised version.
>
>     > e. During the experiment analysis, the results of “Never same” are not presented. Also, there is not much difference, or the difference is not underscored between “environment” and “environment Hard”. This part could be better organized based on the analysis needed.
>
>     We show some results for the `Never Same` environment in the appendix. However, we omit them from the main text. This is due to crowding on the figures. The `Never Same` environment never produces any temporal references as expected, and so the metrics always stay at $0\\%$. We will include the results for this environment in the appendix of the revised version.

---

> > ### Comment · Reviewer_Avm5 · 2023-11-17
> >
> > Thanks for the rebuttal. Most of my questions have been resolved.
> >
> > By "significance between temporally and no temporally specialised messages", I hope to see M metrics of temporal networks statistically outperform the non-temporal networks instead of a direct cutoff of 100%.

---

> ### Author Response · Authors · 2023-11-14
> **Response to Reviewer Avm5 2/2**
>
> ## Questions
>
> 1. > For the temporal loss, I assume it is a categorical classification about the last position the object shows up. I think it could be misleading when trained with target object classification together. There are some circumstances when the objects are the same but the temporal labels are different, which may cause the network to be confused. Have you tried to add a smaller weight to the temporal loss or try to add this loss on top of the temporal LSTM?
>
>    The loss indeed functions as you describe. However, we do not think it should interfere with the object classification as the two losses are computed independently, and use a different branch of the network. We have not experimented with different weighting of this loss, as it is not required for the emergence of temporal references, nor does it improve performance. We have also added this loss to the temporal LSTM, which is the `Temporal` variant, as opposed to the `Temporal-NL` variant which does not feature this loss. We have experimented with using this loss with just the temporal LSTM, removing the meaning LSTM, as we briefly discuss in `Section 3.3, p7`.
>
> 3. > More analysis of the features that the temporal LSTM provides would be helpful to demonstrate its vital role.
>
>    As we touch on in our response to Reviewer `KopV` we can also include metrics regarding the language compositionality in the appendix of the revised version.

---

> ### Author Response · Authors · 2023-11-20
> **Statistical analyses**
>
> Thank you for your continued engagement in the discussion!
>
> As per your comment, we have performed statistical significance analysis, as per below. We confirm that our data for the $M_{\ominus^n}$ for the Temporal and Non-Temporal variants does not come from the same distribution, using the Kolmogorov-Smirnov 2-Sample test. We will include this in the revised version of the paper.
>
> | Metric tested | $p$-value |
> |--------------------------------------------------------------|-----------|
> | $M_{\ominus^1}$                                              | 0.0       |
> | $M_{\ominus^2}$                                              | 0.0       |
> | $M_{\ominus^3}$                                              | 1.16e-220 |
> | $M_{\ominus^4}$                                              | 1.39e-298 |
> | $M_{\ominus^5}$                                              | 1.13e-161 |
> | $M_{\ominus^6}$                                              | 1.46e-144 |
> | $M_{\ominus^7}$                                              | 7.58e-95  |
> | $M_{\ominus^8}$                                              | 1.41e-68  |
>
> We additionally provide detailed statistics regarding the distribution of the $M_{\ominus^n}$ scores, where the value reported is the mean, with standard deviation, min and max values in brackets.
>
> | Agent Type      | $M_{\ominus^1}$                        | $M_{\ominus^2}$                         | $M_{\ominus^3}$                        | $M_{\ominus^4}$                        | $M_{\ominus^5}$                        | $M_{\ominus^6}$                        | $M_{\ominus^7}$                       | $M_{\ominus^8}$                       |
> |-----------------|----------------------------------------|-----------------------------------------|----------------------------------------|----------------------------------------|----------------------------------------|----------------------------------------|---------------------------------------|---------------------------------------|
> | Non-Temporal-NL | 7.80 ($\sigma=17.32$, min=0, max=92.3) | 12.39 ($\sigma=22.10$, min=0, max=88.9) | 5.34 ($\sigma=13.56$, min=0, max=80.0) | 6.23 ($\sigma=14.43$, min=0, max=77.8) | 4.19 ($\sigma=11.50$, min=0, max=72.2) | 3.86 ($\sigma=10.95$, min=0, max=75.0) | 2.77 ($\sigma=9.03$, min=0, max=66.7) | 2.17 ($\sigma=7.95$, min=0, max=66.7) |
> | Non-Temporal    | 7.93 ($\sigma=17.61$, min=0, max=90.9) | 12.88 ($\sigma=22.36$, min=0, max=88.9) | 5.39 ($\sigma=13.56$, min=0, max=76.9) | 6.32 ($\sigma=14.49$, min=0, max=77.8) | 4.28 ($\sigma=11.55$, min=0, max=68.8) | 3.75 ($\sigma=10.73$, min=0, max=69.2) | 2.82 ($\sigma=9.18$, min=0, max=66.7) | 2.15 ($\sigma=7.82$, min=0, max=66.7) |
> | Temporal-NL     | 11.09 ($\sigma=27.80$, min=0, max=100)  | 17.33 ($\sigma=32.90$, min=0, max=100)   | 8.27 ($\sigma=23.65$, min=0, max=100)   | 9.61 ($\sigma=25.10$, min=0, max=100)   | 6.91 ($\sigma=21.70$, min=0, max=100)   | 6.32 ($\sigma=20.78$, min=0, max=100)   | 4.81 ($\sigma=18.32$, min=0, max=100)  | 3.94 ($\sigma=16.64$, min=0, max=100)  |
> | Temporal        | 13.3 ($\sigma=32.61$, min=0, max=100)   | 20.34 ($\sigma=38.12$, min=0, max=100)   | 9.89 ($\sigma=27.98$, min=0, max=100)   | 11.76 ($\sigma=30.03$, min=0, max=100)  | 8.48 ($\sigma=26.05$, min=0, max=100)   | 7.80 ($\sigma=25.02$, min=0, max=100)   | 6.00 ($\sigma=22.16$, min=0, max=100)  | 4.91 ($\sigma=20.18$, min=0, max=100)  |

---

### Author Response · Authors · 2023-11-16
**General Response and Revisions**

We would like to thank all the reviewers for acknowledging that our work is novel and interesting.

We have now updated our manuscript with the changes suggested in the reviews.

We would also like to address two of the common themes emerging from the reviews, and the improvements we have made regarding them:

## Simplicity of the LSTM approach

One common theme is that using just the LSTM is too simplistic of an approach.

We agree that the LSTM is simple, as this was the goal in our study. We present the simplest architecture to show how the temporal references can emerge, given just the ability to understand temporal relationships in the data. The main focus is not the architecture itself, but rather the heuristics that we gain from using a simple architecture. We have further considered both transformers and attention networks, where the attention networks can also develop temporal references. We also show that no additional losses are necessary. We consider this simplicity an advantage of our approach, rather than a negative.

We have amended our paper with the changes to place more emphasis on the results instead of the architecture.

## Notation and examples

The reviewers have pointed out that our notation was not fully consistent across the whole text.
In the revised version, we have improved this, clearing up the notation in the equations, using our LTL-based notation more consistently, and making use of our $M_{\ominus^n}$ notation for the figures.
We have also improved our examples, by using vectors from our definitions instead of single integers, to explain more clearly what they could consist of.
We added more metrics regarding compositionality, and show our results for all environments, in the updated appendices.

---

### Meta-Review · Area_Chair_eaYp · 2023-11-30

**Metareview:**

This paper extends the classic referential game in which two agents have to develop a protocol to successfully communicate about a target object to a temporal domain, in which the objects are ordered. The paper adopts a temporal logic formalism, and designs a special network with a LSTM component to handle temporal information. Several analyses of successful emergence of temporal reference are presented.

The paper offers an original spin on the classic referential game, adding a feature that is obviously important in natural language. However, it is not entirely clear what are the broader implications that it should have. As a contribution to AI, the fact that it relies on an ad-hoc LSTM-based architecture that would certainly not generalize to many tasks severely limits its impact. On the other hand, it's not clear that the ad-hoc environment, the specific architecture and the loss are telling us much about the emergence of temporal reference in natural language.

Thus, while the paper is opening an original research avenue, the authors should further explore how their research fits the broader picture.

**Justification For Why Not Higher Score:**

The setup explored by the authors is too limited and tuned to the task they are interested in to be of general interest to the ICLR audience.

**Justification For Why Not Lower Score:**

N/A

---

### Decision · Program_Chairs · 2024-01-16

Reject